# Topological linear response of hyperbolic Chern insulators

Canon Sun[1,2★], Anffany Chen[1,2], Tomáš Bzdušek[3] and Joseph Maciejko[1,2,4†]

**1** Theoretical Physics Institute, University of Alberta, Edmonton, Alberta T6G 2E1, Canada
**2** Department of Physics, University of Alberta, Edmonton, Alberta T6G 2E1, Canada
**3** Department of Physics, University of Zürich,
Winterthurerstrasse 190, 8057 Zürich, Switzerland
**4** Quantum Horizons Alberta, University of Alberta,
Edmonton, Alberta T6G 2E1, Canada

★ canon@ualberta.ca , † maciejko@ualberta.ca

## Abstract

We establish a connection between the electromagnetic Hall response and band topological invariants in hyperbolic Chern insulators by deriving a hyperbolic analog of the Thouless-Kohmoto-Nightingale-den Nijs (TKNN) formula. By generalizing the Kubo formula to hyperbolic lattices, we show that the Hall conductivity is quantized to $-e^2 C_{ij}/h$, where $C_{ij}$ is the first Chern number. Through a flux-threading argument, we provide an interpretation of the Chern number as a topological invariant in hyperbolic band theory. We demonstrate that, although it receives contributions from both Abelian and non-Abelian Bloch states, the Chern number can be calculated solely from Abelian states, resulting in a tremendous simplification of the topological band theory. Finally, we verify our results numerically by computing various Chern numbers in the hyperbolic Haldane model.

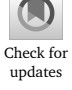

# 1   Introduction

Hyperbolic matter is a novel form of synthetic matter in which particles exist and move on the hyperbolic plane, a two-dimensional space with uniform negative curvature. Simulating physics on the hyperbolic plane in an experimental setting, however, is challenging due to the lack of an isometric embedding of the hyperbolic plane in three-dimensional Euclidean space [1]. This challenge can be overcome by discretizing the hyperbolic plane. By constructing a network with the connectivity of a hyperbolic tight-binding lattice, then as far as the particle is concerned, it is living on a periodic tiling of hyperbolic space. This idea has been realized experimentally in a growing range of platforms, including coplanar waveguide resonators [2, 3], photonic nano/micro-structures [4], mechanical elastic lattices [5], and topoelectric circuits [6–9]. Motivated by these experimental advances, significant theoretical progress has been made to demonstrate the uniqueness and peculiarities of hyperbolic matter [10–44]. Furthermore, hyperbolic lattices can be used to simulate the anti-de Sitter (AdS) space and conformal field theory (CFT) correspondence [45–47] in a laboratory setting [48–56]. Other proposals to simulate negative curvature in condensed matter include Euclidean lattices with non-uniform hoppings [57] and non-Hermitian Hamiltonians [58].

Central to understanding the behavior of electrons in Euclidean lattices is Bloch's theorem. Bloch's theorem states that under translation by a lattice vector $\mathbf{R}$, the electronic wavefunction acquires a phase factor $e^{i\mathbf{k}\cdot\mathbf{R}}$ determined by the crystal momentum $\mathbf{k}$. The crystal momentum $\mathbf{k}$ completely characterizes the symmetry properties of the wavefunction under translations and allows for the classification of energy levels into continuous energy bands. The original statement of Bloch's theorem, however, cannot be straightforwardly applied to hyperbolic lattices. Recently, Ref. [11] generalized Bloch's theorem to hyperbolic lattices, or, more generally, systems with discrete non-Abelian translation groups. The key to this generalization is restating Bloch's theorem in group-theoretic terms: Eigenstates of a Hamiltonian with discrete translational symmetry transform according to irreducible representations (irreps) of the translation group. This non-Abelian Bloch theorem gives rise to two features that are markedly different from the Euclidean Bloch theorem. First, the spectrum admits states transforming under higher-dimensional irreps of the hyperbolic translation group. Bands transforming in a higher-dimensional irrep will have degeneracies protected by the *translation* group, not the point or spin rotation groups. Second, even for one-dimensional irreps, the Brillouin zone (BZ) is in general more complicated. For the {8, 8} lattice studied in Ref. [10], for example (see Fig. 1), the Abelian Brillouin zone is a four-dimensional torus, despite the real-space lattice being a two-dimensional system. The BZ for a hyperbolic lattice is thus far richer than that of an Euclidean lattice.

With a more exotic BZ, the topological band theory of hyperbolic matter becomes more intricate. The central goal of topological band theory is to classify topologically distinct Bloch Hamiltonians. In Euclidean lattices, a comprehensive classification is known based on the dimension and symmetries of the system [59–65], and the corresponding class to which a Hamiltonian belongs can be determined by computing topological invariants, such as the

Chern number [66, 67] or the $\mathbb{Z}_2$ invariant [68, 69]. These invariants are not simply mathematical quantities defined in the abstract but can be observed in physical experiments measuring the charge and/or spin response [66, 68, 70–72]. The situation becomes more complex in hyperbolic lattices. Considering one-dimensional irreps alone, the Abelian BZ is a higher-dimensional torus [10], and thus a multitude of first Chern numbers can be computed, one for each two-dimensional subtorus [13, 14]. Moreover, higher Chern numbers can also be calculated for what is an intrinsically two-dimensional system [9, 16]. For the higher-dimensional irreps, it is not clear how topological invariants can be assigned to them. Indeed, the non-Abelian BZ are complicated moduli spaces [11, 38, 39] that do not in general have a simple toroidal geometry. Furthermore, it is not clear whether any of these topological invariants would be related to physical phenomena of some kind.

In this work, we relate the physical electromagnetic Hall response of translationally invariant hyperbolic insulators to hyperbolic band topological invariants. In Sec. 2, we first briefly review elements of hyperbolic band theory [10, 11]. We consider a general tight-binding Hamiltonian defined on a hyperbolic lattice with periodic boundary conditions and discuss its single-particle spectrum and wavefunctions. In Sec. 3, we define and compute a Hall conductivity using linear response theory (Kubo formula) and show that it is equal to $-e^2 C_{ij}/h$ where $e$ is the electron charge and $C_{ij}$ is a sum of Chern numbers in flux space. The relation between $C_{ij}$ and band invariants is then elucidated and, building on this, we prove that $C_{ij}$ is integer valued, thus demonstrating the quantization of the Hall conductivity. Importantly, we find that *$C_{ij}$ can be computed solely from Abelian Bloch states*, even though all Bloch states (including non-Abelian ones) contribute to the Hall response. This leads to a tremendous simplification of the topological band theory of hyperbolic lattices, since Abelian Bloch states can be characterized analytically. In Sec. 4, we verify those predictions by computing $C_{ij}$ numerically in the hyperbolic Haldane model [13], using finite lattices with periodic boundary conditions that admit both one- and higher-dimensional irreps of the translation group. We conclude briefly in Sec. 5.

## 2 Hyperbolic band theory

The translation group of an arbitrary hyperbolic $\{p, q\}$ lattice (e.g., the $\{8, 8\}$ lattice depicted in Fig. 1) is isomorphic to the fundamental group of a genus-$g$ surface and can be endowed with the presentation

$$\Gamma = \langle \gamma_1, \gamma_2, \dots, \gamma_{2g} | X_g \rangle, \tag{1}$$

with a single relator $X_g$ where each generator $\gamma_j$, $j = 1, \dots, 2g$ and its inverse appear once [12, 14]. The group $\Gamma$ acts on the Poincaré disk $\mathbb{D}$ by fixed-point-free Möbius transformations, and the orbit of the origin $z = 0$ under this action, $L \equiv \{\gamma(0) \in \mathbb{D} | \gamma \in \Gamma\}$, defines a hyperbolic Bravais lattice. In other words, the action of $\Gamma$ partitions $\mathbb{D}$ into disjoint Bravais unit cells, each of which can be labeled by a unique element $z \in L$. In general, the Bravais unit cell contains a basis of $N_s$ sites that can be viewed as sublattice degrees of freedom (e.g., in Fig. 1, $N_s = 1$, while in the hyperbolic Haldane model (Sec. 4), $N_s = 16$). As the action is fixed-point free, there is a one-to-one correspondence between $\Gamma$ and $L$: For each element $\gamma \in \Gamma$, there is a corresponding unit cell $z = \gamma(0) \in L$ and, conversely, for every unit cell $z \in L$ there is a unique $\gamma \in \Gamma$ such that $z = \gamma(0)$. Therefore, we can label unit cells with group elements and vice versa.

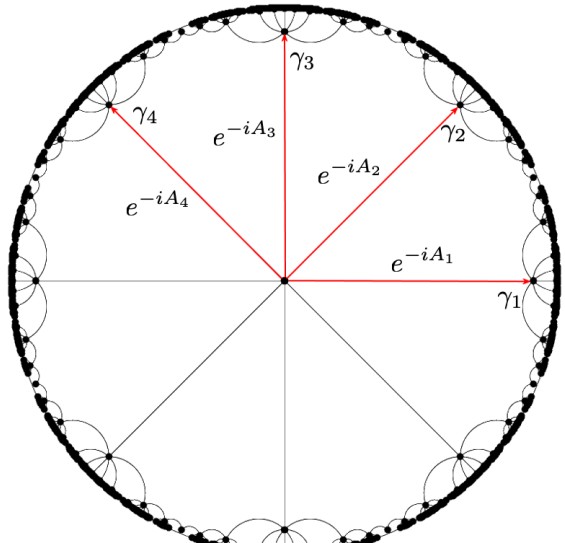

Figure 1: The $\{8,8\}$ tiling of the Poincaré disk. The black dots represent lattice sites and the black lines are geodesics (in the Poincaré metric) connecting nearest neighbors. The red arrows depict the actions of the generators $\gamma_j$ on the lattice site at the origin. In our model, an electron hopping in the $\gamma_j$ direction in the presence of an external gauge field $\mathbf{A}$ acquires a Peierls phase factor $e^{-iA_j}$.

## 2.1 Periodic boundary conditions

To facilitate the discussion of adiabatic charge transport, it is convenient to impose periodic boundary conditions (PBC). The application of PBC amounts to taking the quotient of $\Gamma$ by some normal subgroup $\Gamma_{\mathrm{PBC}}$, whose elements define operations that are "periodic". The quotient group $G \equiv \Gamma/\Gamma_{\mathrm{PBC}}$ of order $|G|$, which we take to be finite, defines a periodic cluster of $|G|$ unit cells [11]. The group $G$ should be interpreted as belonging to a sequence of increasingly large periodic clusters that converges to the thermodynamic limit [35, 36], such that $G$ is meant to approximate the infinite group $\Gamma$. Geometrically, elements of the normal subgroup $\Gamma_{\mathrm{PBC}}$ can be thought of as operations that traverse the entire periodic system and return back to the same unit cell on the periodic cluster. In contrast, the quotient $G$ consists of the residual translations between distinct unit cells on the cluster. While $\Gamma$ is isomorphic to the fundamental group of a genus-$g$ surface, $\Gamma_{\mathrm{PBC}}$ is isomorphic to that of a genus-$h$ surface, with $h \geq g$ scaling with the system size. Put differently, a single unit cell lives on a genus-$g$ surface whereas a periodic cluster lives on a genus-$h$ one. The group $\Gamma_{\mathrm{PBC}}$ can always be given the presentation

$$\Gamma_{\mathrm{PBC}} = \langle \mathfrak{g}_1, \mathfrak{g}_2, \ldots, \mathfrak{g}_{2h} | [\mathfrak{g}_1, \mathfrak{g}_2] \ldots [\mathfrak{g}_{2h-1}, \mathfrak{g}_{2h}] \rangle, \tag{2}$$

where $[a, b] = aba^{-1}b^{-1}$ is the commutator of two group elements. In this presentation, the operation of each generator $\mathfrak{g}_\alpha$, $\alpha = 1, \ldots, 2h$, encircles one of the $2h$ holes of the genus-$h$ surface once. In contrast to $\Gamma$, which is an infinite group, $G$ is a finite (but arbitrarily large) group of order $|G| = (h-1)/(g-1)$ and thus methods from the representation theory of finite groups can be applied.

We consider a nearest-neighbor tight-binding model defined on a periodic cluster $G$ with $N_s$ sublattices, as described by the second-quantized Hamiltonian

$$\hat{H}(\mathbf{A}) = -\sum_{\gamma \in G} \sum_{j=1}^{2g} \sum_{a,b=1}^{N_s} T_{ab}^j e^{-iA_j(\gamma)} \hat{c}_{\gamma\gamma_j,a}^\dagger \hat{c}_{\gamma,b} + \mathrm{h.c.} \tag{3}$$

Here $\hat{c}^{\dagger}_{\gamma,a}$ creates an electron in unit cell $\gamma$ on sublattice $a$. To each link connecting site $b$ in unit cell $\gamma$ to site $a$ in unit cell $\gamma\gamma_j$ is associated a hopping matrix element $T^{j}_{ab}$ and a U(1) Peierls phase factor $e^{-iA_j(\gamma)}$ with connection $A_j(\gamma) \in [0, 2\pi)$. Unlike Euclidean lattices, the connection here is a $2g$-component vector $\mathbf{A} = (A_1, A_2, \ldots, A_{2g})^T$. This hopping model is depicted schematically on the $\{8, 8\}$ lattice in Fig. 1. Note that the nearest neighbor of $\gamma$ by traversing in the $j$ direction is $\gamma\gamma_j$, rather than the perhaps more intuitive $\gamma_j\gamma$. This can be understood in the following way: By definition, $\gamma_j(0)$ is a nearest neighbor of 0 on the Bravais lattice. Translating them both by $\gamma$, we reach the unit cells $\gamma\gamma_j(0)$ and $\gamma(0)$. They must be nearest neighbors of each other because the distance between them is preserved by Möbius maps. All the nearest neighbors of $\gamma(0)$ can be obtained this way. For simplicity, in Eq. (3), we restricted the Hamiltonian to only involve coupling of sites belonging to nearest-neighbor unit cells, but it can be extended to include next-nearest neighbors and more by inserting compatible Peierls phase factors. For example, the next-nearest neighbor hopping from $\gamma$ to $\gamma\gamma_i\gamma_j$ would involve the phase factor $e^{-i[A_j(\gamma\gamma_i)+A_i(\gamma)]}$. In particular, such terms appear in the hyperbolic Haldane model on the $\{8, 3\}$ lattice, studied numerically in Sec. 4.

The connection $A_j(\gamma)$ plays the role of an electromagnetic vector potential applied externally in our model. For any loop around the lattice, the magnetic flux, in units of $\hbar/e$, through the loop is equal to the sum of the individual connections along its path. If the magnetic flux enclosed by any contractible loop is zero, then the connection is flat, meaning there is zero applied magnetic field on the surface. A contractible loop can be defined algebraically as a group element $\gamma \in \Gamma$ that can be reduced to the identity $e \in \Gamma$ using either the trivial relations $\gamma_j\gamma_j^{-1} = \gamma_j^{-1}\gamma_j = e$ or the single non-trivial relation $X_g = e$. On the other hand, non-contractible loops, i.e., those that involve $\mathfrak{g}_\alpha$, can enclose magnetic flux even if the connection is flat. This magnetic flux does not penetrate the surface but instead goes through the $2h$ holes of the surface (see Fig. 2). Mathematically, the periodic cluster resides on a surface with non-trivial homology, which quantifies the number of "holes" the surface possesses. Flat connections corresponding to fluxes threaded through these holes belong to non-trivial cohomological classes.

Generally, the gauge field $A_j(\gamma)$ breaks translational symmetry. The Hamiltonian is only translationally invariant when the gauge field is independent of the position, i.e., $\mathbf{A}(\gamma) = \boldsymbol{\phi}$ for all $\gamma \in G$ and for some $\boldsymbol{\phi} = (\phi_1, \phi_2, \ldots, \phi_{2g})^T$, where $\phi_j \in [0, 2\pi)$. Translationally invariant field configurations are flat because all the relations of $\Gamma$, trivial and non-trivial, contain an equal number of $\gamma_j$ and $\gamma_j^{-1}$, and thus the contribution acquired by traversing through the $\gamma_j$ operation is canceled by that of $\gamma_j^{-1}$. Nevertheless, the gauge field $\phi_j$ can produce magnetic flux through the $2h$ holes (see Fig. 2). Let $\Lambda_j(\gamma)$ be the number of times $\gamma_j$ appears in any word representation of $\gamma \in \Gamma$ minus the number of times $\gamma_j^{-1}$ appears. This is well-defined because all the relations of $\Gamma$ involve the same number of $\gamma_j$ and $\gamma_j^{-1}$. The flux through the hole associated with the generator $\mathfrak{g}_\alpha$, in units of $\hbar/e$, is

$$\varphi_\alpha = \sum_{j=1}^{2g} \Lambda_j(\mathfrak{g}_\alpha)\phi_j. \tag{4}$$

However, in contrast with Euclidean lattices, not all flux configurations $\varphi_\alpha$ can be obtained by a translationally invariant gauge configuration. Here $\Lambda_j(\mathfrak{g}_\alpha)$ can be thought of as a linear map from the space of gauge fields $T^{2g}$, labeled by $\phi_j$, to the space of fluxes $T^{2h}$, $\varphi_\alpha$. The rank of this map is at most $2g$. As the rank is smaller than the dimension of the space of fluxes, $2h$, not all flux configurations are mapped onto by $\Lambda_j(\mathfrak{g}_\alpha)$. To allow for more general flux

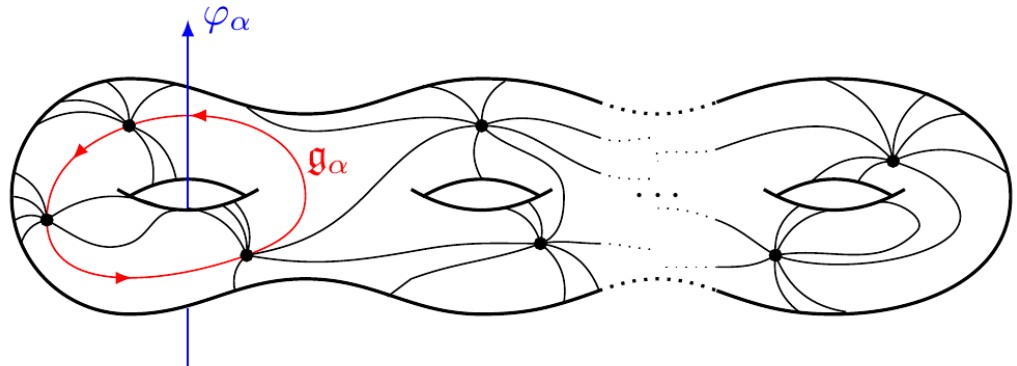

Figure 2: Flux threading on a periodic hyperbolic cluster (black dots representing sites and lines depicting connections), which lives on a higher-genus surface. The non-contractible cycle $\mathfrak{g}_\alpha$ (red) encloses flux $\varphi_\alpha$, which does not penetrate the surface.

configurations, translational symmetry would have to be broken.[1] Since our goal is to relate electromagnetic response coefficients to hyperbolic band invariants, we focus henceforth on translationally invariant gauge configurations.

## 2.2 Single-particle spectrum

As the Hamiltonian in Eq. (3) is non-interacting, the problem of solving for the many-body spectrum reduces to finding the single-particle energy levels. The problem is further simplified when the Hamiltonian is translationally invariant, in which case hyperbolic band theory can be applied [10, 11, 37]. Let $A_j(\boldsymbol{\phi}) = \phi_j$ for all $\gamma \in G$ be a translationally invariant gauge configuration. The single-particle Hilbert space $\mathcal{H} = \ell^2(G) \otimes \mathbb{C}^{N_s}$ is the tensor product of two spaces: $\ell^2(G) \cong \mathbb{C}^{|G|}$, the space of functions on $G$, describing the Bravais lattice, and $\mathbb{C}^{N_s}$, the sublattice degrees of freedom. It is spanned by the orthonormal basis $|\gamma, a\rangle \equiv \hat{c}_{\gamma,a}^\dagger |0\rangle$, which is a position ket at $\gamma$ and in the sublattice $a$. Projecting $\hat{H}$ of Eq. (3) onto $\mathcal{H}$, the single-particle energy spectrum is described by the first quantized Hamiltonian

$$\hat{H}_1(\boldsymbol{\phi}) = -\sum_{\gamma \in G} \sum_{j=1}^{2g} \sum_{a,b=1}^{N_s} T_{ab}^j e^{-i\phi_j} |\gamma\gamma_j, a\rangle \langle \gamma, b| + \text{h.c.} \tag{5}$$

We note that even though the U(1) Peierls phase $e^{-i\phi_j}$ itself is Abelian, the translation symmetry of the Hamiltonian (5) is still noncommutative, described by the non-Abelian group $G$. To make those symmetry properties more manifest, it is advantageous to decompose the Hilbert space into invariant subspaces of $G$. The space $\ell^2(G)$ transforms in the regular representation of $G$, a $|G|$-dimensional representation that is generally reducible, while $\mathbb{C}^{N_s}$ transforms trivially under $G$. The decomposition of the regular representation is achieved through a change of basis for $\mathcal{H}$ from the position basis $|\gamma, a\rangle$ to a new "momentum" basis $|K, \lambda, \nu; a\rangle$ [37]:

$$|\gamma, a\rangle = \sum_{K \in \text{BZ}(G)} \sum_{\lambda, \nu=1}^{d_K} |K, \lambda, \nu; a\rangle \sqrt{\frac{d_K}{|G|}} D_{\nu\lambda}^{(K)}(\gamma), \qquad |K, \lambda, \nu; a\rangle = \sum_{\gamma \in G} |\gamma, a\rangle \sqrt{\frac{d_K}{|G|}} D_{\nu\lambda}^{(K)*}(\gamma). \tag{6}$$

---

[1]One may also consider more general flux configurations that preserve a finite-index normal subgroup of $\Gamma$. Hyperbolic band theory can be applied to the corresponding Bravais supercell [15], which results in an enlarged set of momentum-space Chern numbers [73]. We focus here on the simplest type of electromagnetic Hall response, which relates to Chern invariants defined for the primitive cell.

Here, we denote by BZ($G$) the space of irreps of $G$, by $D^{(K)}(\gamma)$ the unitary representation matrix of $\gamma$ in irrep $K$, and by $d_K$ the dimension of $K$. The index $\nu$ labels the states that mix together under a group transformation and $\lambda$ the copies of the irrep $K$ appearing in the decomposition of $\ell^2(G)$. A feature of the regular representation is that the multiplicity of the irrep $K$ is equal to its dimension $d_K$ [74], thus both $\nu$ and $\lambda$ indices range from 1 to $d_K$. The vector $|K, \lambda, \nu; a\rangle$ transforms in the $K$ irrep of $G$. In this basis, the transformation properties are more transparent. Defining the translation operator $\hat{U}(\gamma)$ by $\hat{U}(\gamma)|\gamma', a\rangle = |\gamma\gamma', a\rangle$, the state $|K, \lambda, \nu; a\rangle$ transforms under the action of $\gamma$ as

$$\hat{U}(\gamma)|K, \lambda, \nu; a\rangle = \sum_{\mu=1}^{d_K} |K, \lambda, \mu; a\rangle D_{\mu\nu}^{(K)}(\gamma). \tag{7}$$

The quantum numbers $K$, $\lambda$, and $a$ define invariant subspaces of $\mathcal{H}$ that are not mixed under hyperbolic translations. The transformation (6) reduces to the change of basis between position and momentum eigenstates when $\Gamma$ is a Euclidean translation group. For illustrative purposes, let us consider the one-dimensional infinite chain with translation group $\Gamma = \mathbb{Z}$ and no sublattice. Imposing PBC corresponds to selecting a normal subgroup $N\mathbb{Z} \triangleleft \mathbb{Z}$ and the quotient $G = \mathbb{Z}_N$ defines a periodic cluster with $|G| = N$ sites [11]. As $G$ is Abelian, all of its irreps are one-dimensional and take the form $D^{(k_n)}(x) = e^{-ik_n x}$, where $x \in \mathbb{Z}_N$ denotes the site and $k_n = 2\pi n/N$, $n = 0, \dots, N-1$, is the crystal wavevector. Making these substitutions, the equations in (6) reduce to the familiar expressions

$$|x\rangle = \frac{1}{\sqrt{N}} \sum_{k_n} |k_n\rangle e^{-ik_n x}, \qquad |k_n\rangle = \frac{1}{\sqrt{N}} \sum_x |x\rangle e^{ik_n x}. \tag{8}$$

The utility of the irrep basis $|K, \lambda, \nu; a\rangle$ is that it block diagonalizes the Hamiltonian. Because of translational symmetry, the Hamiltonian does not mix components that transform under different irreps and takes the form (see App. A for details)

$$\hat{H}_1(\boldsymbol{\phi}) = \sum_{K \in \text{BZ}(G)} \sum_{\lambda, \lambda', \nu, \nu'=1}^{d_K} \sum_{a, a'=1}^{N_s} H_{\lambda\nu a, \lambda'\nu'a'}^{(K)}(\boldsymbol{\phi}) |K, \lambda, \nu; a\rangle\langle K, \lambda', \nu'; a'|, \tag{9}$$

where

$$H_{\lambda\nu a, \lambda'\nu'a'}^{(K)}(\boldsymbol{\phi}) = -\sum_{j=1}^{2g} D_{\lambda'\lambda}^{(K)}(\gamma_j) \delta_{\nu\nu'} T_{aa'}^j e^{-i\phi_j} + \text{h.c.} \tag{10}$$

The Bloch Hamiltonian $H^{(K)}$ is a $d_K^2 N_s \times d_K^2 N_s$ matrix and its spectrum consists of $d_K N_s$ bands, each $d_K$-fold degenerate because of the quantum number $\nu$ (see Fig. 5 for a schematic depiction with $N_s = 3$). Its eigenstates can be brought into "Bloch form". Let $|u_{n\nu}^{(K)}(\boldsymbol{\phi})\rangle$ be a normalized eigenvector of $H^{(K)}(\boldsymbol{\phi})$ with energy $E_n^{(K)}(\boldsymbol{\phi})$, where $n = 1, \dots, d_K N_s$ is a band index. The associated normalized eigenstate to $\hat{H}_1(\boldsymbol{\phi})$ is

$$|\psi_{n\nu}^{(K)}(\boldsymbol{\phi})\rangle = \sum_{\lambda=1}^{d_K} \sum_{a=1}^{N_s} u_{n\nu,\lambda a}^{(K)}(\boldsymbol{\phi}) |K, \lambda, \nu; a\rangle, \tag{11}$$

where $u_{n\nu,\lambda a}^{(K)}(\boldsymbol{\phi}) = \langle K, \lambda, \nu; a|u_{n\nu}^{(K)}(\boldsymbol{\phi})\rangle$. This is a generalization of one variant of Bloch's theorem, which states that, in the Euclidean context, eigenfunctions of a Hamiltonian with discrete translational symmetry are of the form $\psi_n^{(\mathbf{k})}(\mathbf{r}) = u_n^{(\mathbf{k})}(\mathbf{r})e^{i\mathbf{k}\cdot\mathbf{r}}$. Besides the extra index $\nu$ which accounts for the dimension of the irrep, there is also an extra sum over the multiplicity $\lambda$ in Eq. (11). While on Euclidean lattices there is only one basis function per irrep, namely $e^{i\mathbf{k}\cdot\mathbf{r}}$, on hyperbolic lattices, the irrep $K$ has $d_K$ basis functions once $\nu$ is specified. The eigenfunction would generally be a linear combination of all $d_K$ basis functions, which accounts for the extra sum over $\lambda$.

# 3 Hall response on hyperbolic lattices

We now suppose our system is in an insulating state and study its Hall response under an external electric field. We first derive a hyperbolic variant of the Kubo formula based on linear response theory. We then show that the conductivity is related to topological band invariants and is quantized to integer multiples of $e^2/h$.

## 3.1 Hyperbolic Kubo formula

While on Euclidean lattices there are only two independent directions in which a uniform electric field can be applied and current could flow, on hyperbolic lattices this can be generalized to $2g$ possible directions because of the non-commutative nature of the translation group [75]. Suppose the gauge field $A_j$ is varied on top of a stationary, translationally invariant background field $\phi_j$: $A_j(\gamma, t) = \phi_j + \delta A_j(\gamma, t)$. The perturbation $\delta A_j$ generates an electric field $E_j(\gamma, t) = -\delta\dot{A}_j(\gamma, t)\Phi_0/(2\pi)$ along each link. As the link fields $E_j$ are independent, there are $2g$ possible directions to apply an electric field. Similarly, to each generator $\gamma_i$, there is a corresponding charge current operator

$$\hat{J}_i(\gamma) = \frac{2\pi}{\Phi_0}\frac{\partial \hat{H}}{\partial A_i(\gamma)}, \tag{12}$$

which measures the charge current flowing from $\gamma$ to $\gamma\gamma_i$. The $2g$ local currents and electric fields are related, to linear order, through the $2g \times 2g$ conductivity tensor $\sigma_{ij}$:

$$J_i(\gamma, t) = \sum_{j=1}^{2g}\sum_{\gamma' \in G}\int_{-\infty}^{t} dt'\, \sigma_{ij}(\gamma, \gamma'; t-t')E_j(\gamma', t'), \tag{13}$$

where $J_i(\gamma, t) = \langle\hat{J}_i(\gamma, t)\rangle$ is the expectation value of the electric current. As our model is invariant under time translations, it is convenient to switch to the frequency domain, in which case the current-field relation reads

$$J_i(\gamma, \omega) = \sum_{j=1}^{2g}\sum_{\gamma' \in G}\sigma_{ij}(\gamma, \gamma'; \omega)E_j(\gamma', \omega). \tag{14}$$

Our interest is in the direct current (d.c.) conductivity, which can be calculated from the Kubo formula in the $\omega \to 0$ limit (omitting the frequency argument from now on)

$$\sigma_{ij}(\gamma, \gamma') = -i\hbar\sum_{\Omega \neq GS}\frac{\langle GS|\hat{J}_i(\gamma)|\Omega\rangle\langle\Omega|\hat{J}_j(\gamma')|GS\rangle}{(E_\Omega - E_{GS})^2} - (i \leftrightarrow j), \tag{15}$$

where $|\Omega\rangle$ is an eigenstate of the many-body Hamiltonian $\hat{H}$ with energy $E_\Omega$ and $|GS\rangle$ is the ground state with energy $E_{GS}$. In a gapped system, the d.c. and thermodynamic limits commute, thus it is possible to take the $\omega \to 0$ limit first [76, 77].

In the presence of translational symmetry, it is convenient to switch to the Fourier representation. From a group theory perspective, the (discrete) Fourier transform is the decomposition of a function into parts that transform according to particular irreps of the symmetry group $G$. To illustrate this, it is instructive to revisit once again the example of the one-dimensional chain with PBC imposed. The discrete Fourier transform and its inverse on the chain are defined as

$$f(x) = \frac{1}{N}\sum_{k_n}f(k_n)e^{ik_n x}, \qquad f(k_n) = \sum_{x}f(x)e^{-ik_n x}. \tag{16}$$

Here the function $f(x)$ is expanded in terms of the basis functions $D^{(k_n)}(x) = e^{-ik_n x}$, which are themselves the representation "matrices" of the group $G = \mathbb{Z}_N$. Under a translation, basis functions corresponding to different irreps do not mix together. The analogous Fourier transform for non-Abelian groups employs the same idea. Here we simply outline the basic idea and defer the detailed discussion to App. B. By the Peter-Weyl theorem, the representation matrices $D^{(K)*}_{\nu\lambda}(\gamma)$ form a basis for functions on the periodic cluster, thus any function $f$ defined on the cluster can be expanded in terms of them. This motivates the following definition for the Fourier transform on $G$ and its inverse:

$$f(\gamma) = \frac{1}{|G|} \sum_{K \in \mathrm{BZ}(G)} \sum_{\lambda,\nu=1}^{d_K} d_K f^{(K)}_{\lambda\nu} D^{(K)*}_{\nu\lambda}(\gamma), \qquad f^{(K)}_{\lambda\nu} = \sum_{\gamma \in G} f(\gamma) D^{(K)}_{\nu\lambda}(\gamma). \tag{17}$$

Note that because $G$ is non-Abelian, there are extra labels for the Fourier coefficient $f^{(K)}_{\lambda\nu}$, with $\nu$ accounting for the multiple basis functions in the same irrep and $\lambda$ the multiple copies of each irrep. We recover the standard Fourier transform (16) when $G = \mathbb{Z}_N$. As such, (17) is the generalization of the usual Fourier transform from the Euclidean translation group to any finite (but arbitrarily large) non-Abelian group.

The Fourier transform can also be defined for matrix kernels, such as the conductivity tensor Eq. (15). For our purposes, we focus on matrix kernels $h(\gamma, \gamma')$ that are translationally invariant. In other words, $h(\gamma, \gamma') = h(\tilde\gamma\gamma, \tilde\gamma\gamma')$ for all $\tilde\gamma \in G$. Equivalently, $h(\gamma, \gamma') = h(\gamma'^{-1}\gamma)$ is a function of only the "difference variable" $\gamma'^{-1}\gamma$. Since it is a function of one variable, its Fourier transform and inverse are given by

$$h(\gamma'^{-1}\gamma) = \frac{1}{|G|} \sum_{K \in \mathrm{BZ}(G)} \sum_{\lambda,\lambda'=1}^{d_K} d_K h^{(K)}_{\lambda\lambda'} D^{(K)*}_{\lambda'\lambda}(\gamma'^{-1}\gamma), \qquad h^{(K)}_{\lambda\lambda'} = \sum_{\gamma^{-1\prime}\gamma \in G} h(\gamma^{-1\prime}\gamma) D^{(K)}_{\lambda'\lambda}(\gamma^{-1\prime}\gamma). \tag{18}$$

These are the generalizations of the standard Euclidean Fourier and inverse transforms, which for $G = \mathbb{Z}_N$ are

$$h(x - x') = \frac{1}{N} \sum_{k_n} h(k_n) e^{ik_n(x-x')}, \qquad h(k_n) = \sum_{x-x'} h(x - x') e^{-ik_n(x-x')}. \tag{19}$$

Finally, the hyperbolic Fourier transform also exhibits a convolution theorem. Suppose the function $f$ is a convolution of two functions $h$ and $g$, which, in the non-Abelian context, is defined as

$$f(\gamma) = \sum_{\gamma' \in G} h(\gamma'^{-1}\gamma) g(\gamma'). \tag{20}$$

Then the Fourier transform of $f$ is

$$f^{(K)}_{\lambda\nu} = \sum_{\mu=1}^{d_K} h^{(K)}_{\lambda\mu} g^{(K)}_{\mu\nu}, \tag{21}$$

which is simply matrix multiplication. This is the generalization of the Euclidean convolution theorem, which for $G = \mathbb{Z}_N$ states that

$$f(x) = \sum_{x'} h(x - x') g(x') \iff f(k_n) = h(k_n) g(k_n). \tag{22}$$

Returning to the problem of the current response, because the unperturbed Hamiltonian (i.e., with $\delta A_j = 0$) is translationally invariant, so is the conductivity tensor. Using the convolution theorem Eq. (21), the Fourier transform of the current-electric field relation Eq. (14) is

$$J^{(Q)}_{i;\lambda\nu} = \sum_{j=1}^{2g} \sum_{\mu=1}^{d_Q} \sigma^{(Q)}_{ij;\lambda\mu} E^{(Q)}_{j;\mu\nu}. \tag{23}$$

This is the generalization of the relation $J_i(\mathbf{q}) = \sum_{j=1}^d \sigma_{ij}(\mathbf{q})E_j(\mathbf{q})$ in $d$-dimensional Euclidean space. Typically, the Hall measurement is performed in the uniform limit, i.e., $\mathbf{q} \to \mathbf{0}$, which, in group-theoretic terms, is the trivial representation. For this reason, we focus on the response in the trivial representation, denoted "0". Using the Kubo formula, the conductivity tensor in the trivial representation is (see App. C for details)

$$\sigma_{ij}^{(0)}(\boldsymbol{\phi}) = -\frac{e^2}{h}\frac{1}{|G|}\sum_{K \in \mathrm{BZ}(G)} 2\pi F_{ij}^{(K)}(\boldsymbol{\phi}),\tag{24}$$

where $F_{ij}^{(K)} \equiv i\sum_{n<0}\sum_{\nu=1}^{d_K}\langle\partial_{\phi_i}u_{n\nu}^{(K)}|\partial_{\phi_j}u_{n\nu}^{(K)}\rangle - (i \leftrightarrow j)$ is the Berry curvature in flux space associated with the irrep $K$. As the effect of the gauge fields $\phi_i$ is to simply change the boundary conditions, assuming the bulk conductivity is insensitive to the boundary conditions, the Berry curvature itself is expected to be quantized [78,79]. This allows us to average over the gauge fields, yielding

$$\sigma_{ij}^{(0)} \equiv \left\langle \sigma_{ij}^{(0)}(\boldsymbol{\phi}) \right\rangle_{\boldsymbol{\phi}} = -\frac{e^2}{h}C_{ij},\tag{25}$$

where

$$C_{ij} = \frac{1}{|G|}\sum_{K \in \mathrm{BZ}(G)} C_{ij}^{(K)}, \qquad C_{ij}^{(K)} = \frac{1}{2\pi}\int_0^{2\pi} d\phi_i d\phi_j F_{ij}^{(K)}.\tag{26}$$

Eq. (25) is the generalization of the Niu-Thouless-Wu (NTW) formula to hyperbolic lattices. Here $C_{ij}^{(K)}$ is the first Chern number associated with the irrep $K$ and is quantized to integer values. Although not obvious from Eq. (26), we will prove in Sec. 3.3 that $C_{ij}$ is also an integer, which implies the Hall conductivity is quantized.

## 3.2 Band invariants

In systems exhibiting translational symmetry, the Chern number $C_{ij}$ is related to band topological invariants. This is well-known in Euclidean lattices, in which $C_{ij}$ is the surface integral of the Berry curvature *in momentum space* over the BZ [66]. The key insight lies in the fact that a change in $\boldsymbol{\phi}$ is equivalent to a change in the crystal momentum $\mathbf{k}$ of the Bloch state $|u_{\mathbf{k}}\rangle$. Put differently, changing $\boldsymbol{\phi}$ amounts to parallel transporting $|u_{\mathbf{k}}\rangle$ around the BZ. This correspondence between $\boldsymbol{\phi}$ and $\mathbf{k}$ allows the integral over fluxes in Eq. (26) to be recast into an integral over the BZ.

To generalize to the hyperbolic case, it is useful to rephrase the correspondence between $\boldsymbol{\phi}$ and $\mathbf{k}$ in group-theoretic terms. The momentum $\mathbf{k}$ defines an irrep of the Euclidean translation group and the BZ is the collection of irreps. The flow induced by $\boldsymbol{\phi}$ can be thought of as a flow in the space of irreps. Crucially, this interpretation remains unchanged in the hyperbolic case. In this section, we demonstrate that adiabatically changing the flux $\boldsymbol{\phi}$ leads to a flow in the space of irreps of the hyperbolic translation group, subsequently establishing a correspondence between $C_{ij}$ and band topological invariants.

First, we extend the domain of the Bloch Hamiltonian $H^{(K)}$ from $\mathrm{BZ}(G)$ to $\mathrm{BZ}(\Gamma)$, the space of irreps of $\Gamma$ [11,37–39]. This is achieved by simply allowing the representation matrices $D^{(K)}$ in Eq. (10) to be unitary irreps of $\Gamma$. The domain has been enlarged because an irrep of $G$ is also one of $\Gamma$. To be more precise, an irrep $K \in G$ can be lifted to an irrep $\hat{K}$ of $\Gamma$ by defining

$$D^{(\hat{K})}(\gamma) \equiv D^{(K)}([\gamma]), \qquad \forall \gamma \in \Gamma,\tag{27}$$

where $[\gamma] \in G = \Gamma/\Gamma_{\mathrm{PBC}}$ is the coset to which $\gamma$ belongs. $\hat{K}$ is a representation because the cosets themselves form a group:

$$D^{(\hat{K})}(\gamma_1\gamma_2) \equiv D^{(K)}([\gamma_1\gamma_2]) = D^{(K)}([\gamma_1][\gamma_2]) = D^{(K)}([\gamma_1])D^{(K)}([\gamma_2]) = D^{(\hat{K})}(\gamma_1)D^{(\hat{K})}(\gamma_2).\tag{28}$$

It is irreducible because it consists of all the representation matrices of $K$, and $K$ is irreducible. Hence, BZ($G$) is a subset of BZ($\Gamma$).

For illustrative purposes, it is instructive to consider the one-dimensional chain again. The irreps of $G = \mathbb{Z}_N = \mathbb{Z}/N\mathbb{Z}$ are $D^{(k_n)}([x]) = e^{-ik_n[x]}$, where $[x] \in G$ with the equivalence relation $x \sim y$ iff $x$ and $y$ differ by an integer multiple of $N$. The wavevector satisfies the quantization condition $k_n = 2\pi n/N$, $n = 0,\ldots,N-1$, to ensure $D^{(k_n)}$ is independent of the representative chosen from the coset. The irrep $k_n$ can be lifted to an irrep $\hat{k}_n$ of $\Gamma = \mathbb{Z}$ by defining $D^{(\hat{k}_n)}(x) = e^{-ik_n x}$, where $x \in \Gamma$. The irreps $\hat{k}_n$ constitute a subset of BZ($\Gamma$) because the irreps of $\Gamma$ are of the form $D^{(k)}(x) = e^{-ikx}$, where $k \in [0, 2\pi)$ with no quantization condition imposed. This lifting procedure allows us to regard irreps of $G$ also as those of $\Gamma$ and to define the Bloch Hamiltonian over BZ($\Gamma$).

We now discuss the effect of the gauge field $\boldsymbol{\phi}$ on the Bloch Hamiltonian $H^{(K)}(\boldsymbol{\phi})$ and eigenstates $|u_{n\nu}^{(K)}(\boldsymbol{\phi})\rangle$. Consider the combination $D^{(K')}(\gamma_j) \equiv e^{-i\phi_j} D^{(K)}(\gamma_j)$ in the Bloch Hamiltonian in Eq. (10). While the map $D^{(K')}$ is defined over the generators of $\Gamma$, it can be extended to all of $\Gamma$ straightforwardly by defining $D^{(K')}(\gamma) \equiv \chi^{(\boldsymbol{\phi})}(\gamma) D^{(K)}(\gamma)$ for all $\gamma \in \Gamma$, where $\chi^{(\boldsymbol{\phi})}(\gamma) = e^{-i\sum_{j=1}^{2g} \phi_j \Lambda_j(\gamma)}$. The matrices $D^{(K')}(\gamma)$ for all $\gamma \in \Gamma$ furnish an irrep of $\Gamma$. To see this, notice that $\chi^{(\boldsymbol{\phi})}$ is a one-dimensional representation of $\Gamma$. This makes $D^{(K')}$ the tensor product representation $\chi^{(\boldsymbol{\phi})} \otimes D^{(K)}$. As taking the tensor product of an irrep with a one-dimensional irrep yields another irrep, $K'$ is an irrep of $\Gamma$ (with the same dimension, $d_{K'} = d_K$). Since this is true for all $\boldsymbol{\phi}$, we can regard $K$ as a continuous function of $\boldsymbol{\phi}$ by defining $K(\boldsymbol{\phi}) = K'$, and write

$$H^{(K(\boldsymbol{\phi}))}(0) = H^{(K(0))}(\boldsymbol{\phi}).\tag{29}$$

Note that while $D^{(K(\boldsymbol{\phi}))}$ is a linear representation of $\Gamma$, it is generally not one of $G$, as $D^{(K(\boldsymbol{\phi}))}(\mathfrak{g}_\alpha) = e^{-i\varphi_\alpha} D^{(K(\boldsymbol{\phi}))}(e)$, which has to equal $D^{(K(\boldsymbol{\phi}))}(e)$ for it to be an irrep of $G$. Therefore, $K(\boldsymbol{\phi})$ is an irrep of $G$ if and only if the fluxes through all $2h$ handles are integer multiples of the flux quantum. As for the Bloch states, Eq. (29) implies that

$$|u_{n\nu}^{(K(\boldsymbol{\phi}))}(0)\rangle = |u_{n\nu}^{(K(0))}(\boldsymbol{\phi})\rangle.\tag{30}$$

Thus, the insertion of flux leads to a flow in BZ($\Gamma$). Since $K(\boldsymbol{\phi})$ has the same dimension as $K(0)$, the flow does not change the dimension of the irrep. Thus, for each $K$ in Eq. (26), this flow takes place inside a single component of BZ($\Gamma$), which is a moduli space of flat $U(d_K)$ connections (or equivalently, vector bundles of rank $d_K$) over the genus-$g$ Riemann surface $\mathbb{D}/\Gamma$ [11, 37–39].

We can now relate the Chern number to band invariants. To do this, we recast the flux integrals in $C_{ij}^{(K)}$ into a surface integral over some region in the hyperbolic BZ. For each irrep $K$ of $G$, define the surface $S_{ij}^{(K)} = \{K(\boldsymbol{\phi}) \in \mathrm{BZ}(\Gamma) | 0 \le \phi_i, \phi_j < 2\pi\}$, which is the surface traced out in BZ($\Gamma$) when $\phi_i$ and $\phi_j$ are varied continuously, starting at $K(0) \in \mathrm{BZ}(G)$ (see Fig. 3). The map $K$ defines a one-to-one correspondence between points on the flux torus $T^2 \ni (\phi_i, \phi_j)$ and the surface $S_{ij}^{(K)}$. The map is injective because the representation matrices for the generators, $D^{(K(\boldsymbol{\phi}))}(\gamma_j) = e^{-i\phi_j} D^{(K(0))}(\gamma_j)$, are different for different $\phi_j \in (0, 2\pi]$; it is surjective because $S_{ij}^{(K)}$ is defined as the image of $K$. Since the space BZ($\Gamma$) of $d_K$-dimensional irreps is a smooth manifold [38, 80], the map $K : T^2 \to S_{ij}^{(K)}$ is a diffeomorphism. Therefore, by a change of variables we can express the integral as one over the coordinates of $S_{ij}^{(K)}$. This is more compactly written in the notation of differential forms as

$$C_{ij}^{(K)} = \frac{1}{2\pi} \int_{S_{ij}^{(K)}} (K^{-1})^* F^{(K)},\tag{31}$$

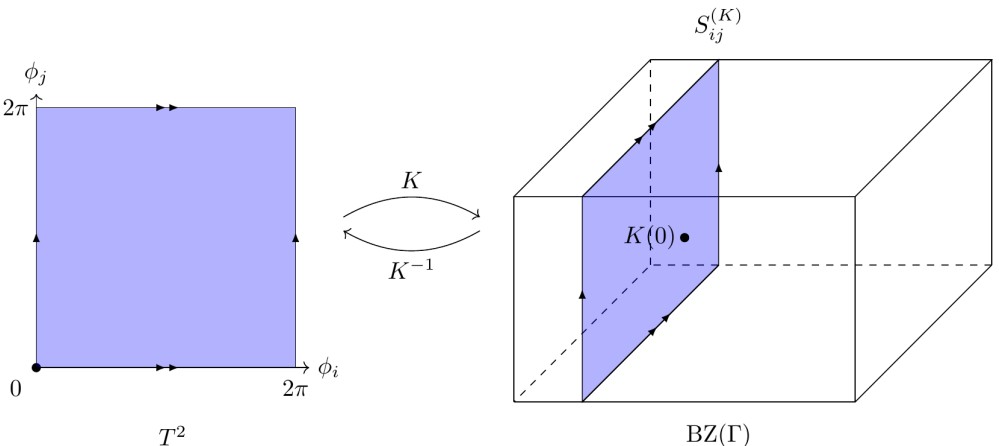

Figure 3: Under the map $K$, the flux torus $T^2$ (left, blue) gets mapped to the surface $S_{ij}^{(K)}$ (right, blue) in the hyperbolic BZ (schematically drawn as a cube). The point $K(0)$ lies on the surface and is an irrep of $G$.

where $(K^{-1})^* F^{(K)}$ is the pullback of the Berry curvature $F^{(K)}$ under the map $K^{-1} : S_{ij}^{(K)} \to T^2$. In this form, the Chern number can be interpreted as the integral of the Berry curvature in the hyperbolic Brillouin zone over the closed surface $S_{ij}^{(K)}$, which is a hyperbolic analog of the Thouless-Kohmoto-Nightingale-den Nijs (TKNN) formula [66].

A more explicit formula can be given for $C_{ij}^{(K)}$ when $K$ is a one-dimensional irrep. The one-dimensional irreps of $G$ are labeled by a $2g$-component crystal wavevector $\mathbf{k}$ that lives on a $2g$-dimensional torus and satisfies the quantization condition

$$2\pi n_\alpha = \sum_{j=1}^{2g} \Lambda_j(\mathfrak{g}_\alpha) k_j \,, \tag{32}$$

for some $n_\alpha \in \mathbb{Z}$ [10, 11]. The generators are represented as $D^{(\mathbf{k})}(\gamma_j) = e^{-ik_j}$. When the external gauge field is applied, the wavevector shifts to $\mathbf{k}(\phi) = \mathbf{k} + \phi$. Performing a change of variables, we obtain the Chern number associated with the irrep $\mathbf{k}_0 \in \mathrm{BZ}(G)$ to be

$$C_{ij}^{(\mathbf{k}_0)} = \frac{1}{2\pi} \int \mathrm{d}k_i \mathrm{d}k_j F_{ij}(\mathbf{k}_0 + \mathbf{k}) \,, \tag{33}$$

where $F_{ij}(\mathbf{k}) = i \sum_{n<0} \langle \partial_{k_i} u_n^{(\mathbf{k})} | \partial_{k_j} u_n^{(\mathbf{k})} \rangle - (i \leftrightarrow j)$ is the Berry curvature in momentum space. In other words, $C_{ij}^{(\mathbf{k}_0)}$ is the Chern number associated with the subtorus $(k_i, k_j)$ of $T^{2g}$. This Chern number has been computed in various models of hyperbolic Chern insulators, such as in Refs. [13, 14]. Furthermore, when $g = 1$, e.g., the square lattice, $C_{ij}^{(\mathbf{k}_0)}$ reduces to the usual Chern number for Euclidean lattices.

## 3.3 Quantization of the Hall conductivity

We now show that the Hall conductivity is quantized. This follows because the Chern numbers $C_{ij}^{(K)}$ for all $K \in \mathrm{BZ}(G)$ are not independent. Remarkably, as we prove below,

$$C_{ij}^{(K)} = d_K^2 C_{ij}^{(0)} \,, \tag{34}$$

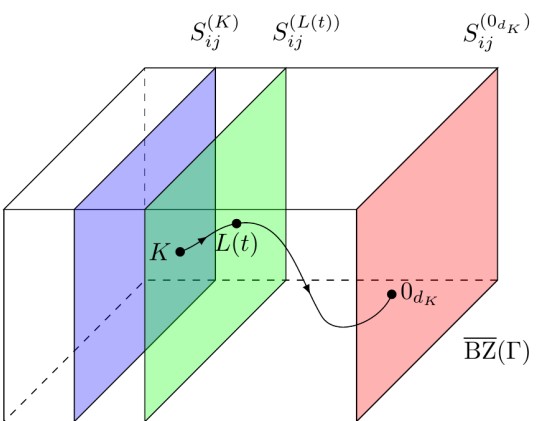

Figure 4: Deforming the surface $S_{ij}^{(K)}$ into $S_{ij}^{(0_{d_K})}$. The cuboid is a schematic representation of $\overline{\text{BZ}}(\Gamma)$, the space of all representations of $\Gamma$ (reducible and irreducible). Through flux threading, $K$ traces out the surface $S_{ij}^{(K)}$ (blue) and $0_{d_K}$ traces out $S_{ij}^{(0_{d_K})}$ (red). As the BZ is connected, $K$ and $0_{d_K}$ are connected by a path $L(t)$, and at each point along the path, there is a corresponding smooth surface $S_{ij}^{(L(t))}$ (green).

where $C_{ij}^{(0)}$ is the Chern number associated with the trivial representation. Equation (34) implies $C_{ij}$ is an integer, because [using Eq. (26)]:

$$C_{ij} \equiv \frac{1}{|G|} \sum_{K \in \text{BZ}(G)} C_{ij}^{(K)} = \frac{1}{|G|} \sum_{K \in \text{BZ}(G)} d_K^2 C_{ij}^{(0)} = C_{ij}^{(0)} \in \mathbb{Z}. \tag{35}$$

In the last step, we have used that $\sum_{K \in \text{BZ}(G)} d_K^2 = |G|$. Therefore, the Hall conductivity in Eq. (25) becomes

$$\sigma_{ij}^{(0)} = -\frac{e^2}{h} C_{ij}^{(0)}. \tag{36}$$

Eq. (36) marks the central result of this work. The Hall conductivity on a hyperbolic lattice is quantized to integer multiples of $e^2/h$, with the integer multiple being the Chern number associated with the trivial irrep. Importantly, it is independent of the choice of periodic cluster $G$. This is unlike, for example, the density of states, where an appropriate sequence of periodic clusters is needed to reach the correct result in the thermodynamic limit [15,35,36]. Furthermore, even though all the irreps contribute to the response, to determine the conductivity it suffices to compute $C_{ij}^{(0)}$ using Eq. (33). Since the trivial irrep is one-dimensional, this can be accomplished using Abelian hyperbolic band theory [10], which makes it considerably simpler than evaluating $C_{ij}$ directly by summing over all (Abelian and non-Abelian) irreps.

We now prove Eq. (34). The idea is to deform the surface $S_{ij}^{(K)}$ continuously to a different surface that allows us to easily relate the Chern numbers. To that end, let $K$ be a $d_K$-dimensional irrep of $G$ and $0_{d_K}$ the representation that is the direct sum of $d_K$ copies of the trivial representation. We next rely on two properties of representations of $\Gamma$, namely that the space $\text{BZ}(\Gamma)$ of $d_K$-dimensional irreps is a smooth and connected manifold [38,80] and the space $\overline{\text{BZ}}(\Gamma)$ of all $d_K$-dimensional representations (whether reducible or irreducible) is connected [81]. Owing to the listed properties, there exists a continuous path $L(t)$, $0 \leq t \leq 1$, that starts at $L(0) = K$ and ends at $L(1) = 0_{d_K}$ (see Fig. 4), such that $L(t)$ is irreducible and thereby smooth for all $t < 1$. At each point $L(t)$, we can consider as in Sec. 3.2 the surface $S_{ij}^{(L(t))}$ traced out by $L(t)$ under flux threading. By assumption of our system being an insulator, the gap does not close at any points along $L(t)$, and thus $S_{ij}^{(K)}$ can be smoothly deformed into $S_{ij}^{(0_{d_K})}$. As the

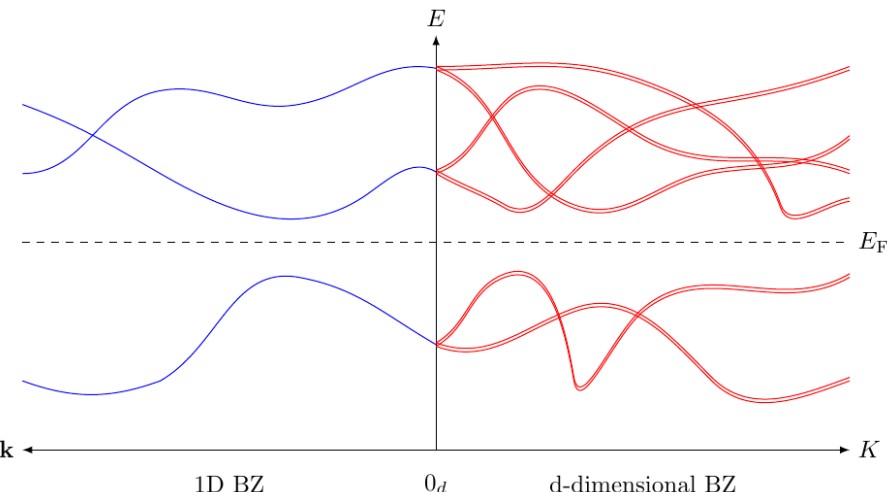

Figure 5: Because the BZ for irreps of a given dimension $d$ is connected, the filling fraction $f_d$ of the associated bands for a hyperbolic insulator is the same for all $d$ (here $f_d = 1/3$, with $d = 1$ bands in blue and $d = 2$ bands in red).

Chern number is invariant under smooth deformations, we have $C_{ij}^{(K)} = C_{ij}^{(0_{d_K})}$. Eq. (34) follows once we show that $C_{ij}^{(0_{d_K})} = d_K^2 C_{ij}^{(0)}$. Indeed, the Bloch Hamiltonian for the representation $0_{d_K}$ is

$$
\begin{aligned}
H_{\lambda\nu a;\lambda'\nu'a'}^{(0_{d_K})} &= -\sum_{j=1}^{2g} \delta_{\lambda'\lambda}\delta_{\nu\nu'}T_{aa'}^j e^{-i\phi_j} + \text{h.c.} \\
&= \delta_{\lambda'\lambda}\delta_{\nu\nu'}H_{aa'}^{(0)}.
\end{aligned}
\tag{37}
$$

This Hamiltonian is simply the direct sum of $d_K^2$ copies of the Bloch Hamiltonian associated with the trivial representation $H^{(0)}$. As the Chern number is additive under direct sum, we have $C_{ij}^{(0_{d_K})} = d_K^2 C_{ij}^{(0)}$ as required.

The connectedness of the BZ in a given irrep dimension reveals another generic feature of insulators in hyperbolic space: The filling fraction of U($d$) bands, i.e., bands with a $d$-fold degeneracy at every $K$ that is protected by translation symmetry, is the same for all $d$ (Fig. 5). This phenomenon was observed previously in hyperbolic tight-binding models with flat bands [25] but not given an explanation. To understand this, let $f_1$ be the fraction of filled U(1) bands ($N_s = 3$ and $f_1 = 1/3$ in the schematic example of Fig. 5). Diagonalizing the Bloch Hamiltonian in Eq. (37) at $0_d$ in the U($d$) BZ, the number of filled states is $f_1 d^2 N_s$. As $K$ continuously moves away from $0_d$, the energy levels form bands that, by assumption, do not cross the Fermi level, so the number of filled states for each $K$ remains $f_1 d^2 N_s$. Therefore, the filling fraction of U($d$) bands is

$$
f_d = \frac{\text{number of filled U}(d)\text{ states at each }K}{\text{number of U}(d)\text{ states at each }K} = \frac{f_1 d^2 N_s}{d^2 N_s} = f_1 \equiv f \,.
\tag{38}
$$

Note that when moving away from $0_d$, the degeneracy from the $\lambda$ quantum number is generically lifted but that from $\nu$ is not. Hence, there are generically $f d N_s$ number of filled U($d$) bands. This argument implies bands from various dimensions are not independent but are connected in the spectrum to form a "wider band". Put differently, a band in a hyperbolic electronic system consists of $d$ number of U($d$) subbands, for each $d \geq 1$.

Table 1: The two independent Chern numbers of the hyperbolic Haldane model on the $\{8,3\}$ lattice, computed at various filling fractions $f$ for the parameters adopted in Ref. [13].

| $f$ | $C_{12}$ | $C_{13}$ |
|---|---|---|
| 5/16 | $-1$ | 1 |
| 1/2 | 0 | 0 |
| 11/16 | $-1$ | 1 |

# 4 Chern numbers in the hyperbolic Haldane model

To supplement our mathematical arguments, we numerically computed the flux-space Chern numbers $C_{ij}$ of the $\{8,3\}$ hyperbolic Haldane model [13,14] on periodic clusters with $|G|$ unit cells ($N_s|G|$ sites). We here only outline the basic steps of the computation; supplementary data and code that can be used to reproduce our results are available at Ref. [82]. We performed computations on a single Abelian cluster with 20 unit cells (320 sites), and four non-Abelian clusters with $\{24, 48, 56, 100\}$ unit cells ($\{384, 768, 896, 1600\}$ sites). A periodic cluster is termed Abelian (non-Abelian) if the finite-size translation group $G = \Gamma/\Gamma_{\mathrm{PBC}}$ is Abelian (non-Abelian) [11]. We use the same model parameters as in Ref. [13], i.e., next-nearest-neighbor hopping amplitude $t_2 = 1/6$ and sublattice mass $M = 1/3$ (in units of the nearest-neighbor hopping amplitude), and next-nearest-neighbor hopping phase $\Phi = \pi/2$. The model is defined on the $\{8,8\}$ Bravais lattice which possesses 4 translation generators and, consequently, has $\binom{4}{2} = 6$ Chern numbers. As shown in Ref. [14] however, for $M \neq 0$ only two of the Chern numbers—$C_{12}$ and $C_{13}$—are independent due to point-group symmetry constraints. For our choice of parameters, the hyperbolic Haldane model exhibits three energy gaps,[2] namely at filling fractions $f \in \{5/16, 1/2, 11/16\}$, with the first and the last being topological. We discretized the flux integral in Eq. (26), diagonalized the Hamiltonian at every flux point, and computed the Chern number using the method described in Ref. [83]. The results are summarized in Table 1. We find that, for the clusters studied, $C_{ij}$ is always an integer and independent of the cluster used. Crucially, we also find they are equal to the momentum-space Chern numbers evaluated in Ref. [13], which verifies our prediction in Eq. (35).

Furthermore, we computed the Chern numbers $C_{ij}^{(K)}$ for each irrep of two non-Abelian periodic clusters and verified Eq. (34). The first cluster has order $|G| = 24$ and comprises eight one-dimensional irreps and four two-dimensional irreps. The second has order $|G| = 48$ and is composed of eight one-dimensional irreps, six two-dimensional irreps, and a single four-dimensional irrep. Detailed information about the two clusters are available in Ref. [82]. To evaluate $C_{ij}^{(K)}$, we isolated states transforming under the irrep $K$ from the spectrum by introducing the projection operators [11]

$$\hat{\Pi}^{(K)} = \frac{d_K}{|G|} \sum_{\gamma \in G} \chi^{(K)*}(\gamma) \hat{U}(\gamma), \tag{39}$$

where $\chi^{(K)}(\gamma)$ is the character of $\gamma$ in the irrep $K$. The projector $\hat{\Pi}^{(K)}$, when applied to an arbitrary state, selects out the part of the state that transforms in the irrep $K$. As the projectors commute with the Hamiltonian and with one another, they can be simultaneously diagonalized

---

[2]Note that for the continuous deformation arguments in Sec. 3.3 to apply, the gaps must persist in the thermodynamic limit and for irreps of all dimensions, which is not trivial to verify for hyperbolic lattices. For example, the $\{8,8\}$ lattice Dirac model [9] is gapped at the level of one-dimensional irreps but becomes gapless once higher-dimensional irreps are taken into account, for a certain range of parameters [16]. A study using the supercell method [15] suggests that the gaps in the $\{8,3\}$ Haldane model first identified in Ref. [13] do persist in the thermodynamic limit.

alongside the Hamiltonian. We identified eigenstates transforming in the irrep $K$ by isolating those with eigenvalue of $\hat{\Pi}^{(K)}$ equal to 1. Subsequently, we computed the Chern numbers $C_{ij}^{(K)}$ from these eigenstates, confirming Eq. (34) for both clusters studied. Moreover, we verified that the filling fraction $f_d$ is indeed independent of $d$, as expected from Eq. (38).

## 5 Summary and outlook

In conclusion, we established a connection between the electromagnetic Hall response of hyperbolic lattice insulators and their topological invariants from hyperbolic band theory—that is, we derived a hyperbolic analog of the TKNN formula. By doing so, we elucidated the physical meaning of the momentum-space Chern numbers computed in, for example, Ref. [13,14]. Furthermore, we demonstrated that both Abelian and non-Abelian Bloch states contribute to the Hall response. However, to compute the Hall conductivity, we have shown that it is sufficient to consider Abelian Bloch states, which tremendously simplifies the structure of topological band theory for hyperbolic lattices. For this result to hold, the spectrum must be truly insulating, i.e., fully gapped for Bloch states of arbitrary irrep dimension $d \geq 1$.

In Sec. 3, we derived the Hall conductivity based on the Kubo formalism and related it to band invariants. We first showed that each irrep has an associated Chern number, defined through a flux-space integral. The Hall conductivity is the sum of all these Chern numbers divided by the number of unit cells in the periodic cluster. We then converted each Chern number into a band invariant by demonstrating that an adiabatic insertion of the flux leads to a flow in the hyperbolic Brillouin zone. The Chern numbers for irreps of various dimensions were shown to be interrelated in a simple manner, combining to produce a quantized Hall response. Finally, we explicitly verified our theoretical predictions through numerical computations of Chern numbers in the $\{8, 3\}$ hyperbolic Haldane model.

Beyond numerical simulations, our theoretical predictions can in principle be verified experimentally using existing techniques. Ref. [3] utilized non-reciprocal tunable phase shifters in a microwave-frequency scattering network to perform a flux-insertion experiment in a hyperbolic lattice with Corbino disk geometry. Those techniques could potentially be adapted for a measurement of the Hall response discussed here. In Ref. [8], a hyperbolic topoelectric circuit was realized in the geometry of a periodic cluster, using tunable complex-phase elements that could in principle be used to both engineer the complex hoppings needed for a Chern insulator model, and also to perform flux insertion via the tunable Peierls phases in Eq. (3). Beyond the Chern number, our work opens the door to a systematic generalization of Euclidean topological band theory to hyperbolic lattices. For example, it would be interesting to investigate whether other types of band invariants, such as the $\mathbb{Z}_2$ invariant of two-dimensional time-reversal invariant insulators [68,69], can likewise be shown to be controlled by the momentum-space topology of Abelian Bloch states. It would also be valuable to elucidate any possible relationship between the Chern numbers defined here and the invariants from noncommutative geometry introduced by Mathai and co-workers for the quantum Hall effect on the hyperbolic plane [75,84–87]. We finally remark that while we focused on the Hall response in this work, the momentum-space methods we introduced, based on the generalized Fourier transform in Eq. (17), are very general and can be adapted to other kinds of many-body or linear-response calculations on hyperbolic lattices. In App. C, as part of our calculation of the Hall conductivity, we derived the generalization of the Fourier transform of the current operator for hyperbolic lattices. We also showed that the hyperbolic analog of the addition of crystal momentum is the tensor product of irreps. These methods can pave the way for future theoretical research in hyperbolic lattices.

## Acknowledgments

The authors thank Albion Arifi, Igor Boettcher, Santanu Dey, Bastian Heß, Davidson Noby Joseph, Steven Rayan, G. Shankar, and Mireia Tolosa-Simeón for fruitful discussions. The numerical computation was enabled in part by support provided by Compute Ontario (computeontario.ca) and the Digital Research Alliance of Canada (alliancecan.ca).

**Funding information**  C.S. acknowledges support through the Natural Sciences and Engineering Research Council of Canada (NSERC) Discovery Grant RGPAS-2020-00064 and the Pacific Institute for the Mathematical Sciences CRG PDF Fellowship Award. A.C. acknowledges the support of NSERC Discovery Grant RGPIN-2020-06999, Avadh Bhatia Fellowship, startup fund UOFAB Startup Boettcher, and the Faculty of Science at the University of Alberta. T.B. was supported by the Starting Grant No. 211310 by the Swiss National Science Foundation (SNSF). J.M. was supported by NSERC Discovery Grants RGPIN-2020-06999 and RGPAS-2020-00064; the Canada Research Chair (CRC) Program; and Alberta Innovates.

## A  Bloch Hamiltonian

In this appendix, we write the first-quantized Hamiltonian from Eq. (5) in block-diagonalized form [Eq. (9)]. The Hamiltonian can be brought to block diagonal form based purely on symmetry constraints. The idea is that because of translational symmetry, the Hamiltonian commutes with all hyperbolic translations, and thus by Schur's lemma, the Hamiltonian cannot mix parts of the Hilbert space transforming under different irreps.

The Hamiltonian is automatically block diagonalized once expressed in the basis $|K, \lambda, \nu; a\rangle$. In other words, we write the Hamiltonian as

$$\hat{H}_1 = \sum_{K,K' \in \mathrm{BZ}(G)} \sum_{\lambda,\nu=1}^{d_K} \sum_{\lambda',\nu'=1}^{d_{K'}} \sum_{a,a'=1}^{N_s} \langle K, \lambda, \nu; a | \hat{H}_1 | K', \lambda', \nu'; a' \rangle | K, \lambda, \nu; a \rangle \langle K', \lambda', \nu'; a' |. \quad (A.1)$$

The matrix element $\langle K, \lambda, \nu; a | \hat{H}_1 | K', \lambda', \nu'; a' \rangle$ can be evaluated by inserting the resolution of the identity $\mathbb{1}_{\mathcal{H}} = \sum_{\gamma \in G} \sum_{b=1}^{N_s} | \gamma, b \rangle \langle \gamma, b |$ and using the transformation (6),

$$
\begin{aligned}
\langle K, \lambda, \nu; a | \hat{H}_1 | K', \lambda', \nu'; a' \rangle &= \sum_{\gamma,\gamma' \in G} \sum_{b,b'=1}^{N_s} \langle K, \lambda, \nu; a | \gamma, b \rangle \langle \gamma, b | \hat{H}_1 | \gamma', b' \rangle \langle \gamma', b' | K', \lambda', \nu', a' \rangle \\
&= -\frac{\sqrt{d_K d_{K'}}}{|G|} \sum_{\gamma \in G} \sum_{j=1}^{2g} D_{\nu\lambda}^{(K)}(\gamma\gamma_j) T_{aa'}^j e^{-i\phi_j} D_{\nu'\lambda'}^{(K')*}(\gamma) + \mathrm{h.c.} \\
&= -\frac{\sqrt{d_K d_{K'}}}{|G|} \sum_{\gamma \in G} \sum_{j=1}^{2g} \sum_{\mu=1}^{d_K} D_{\nu\mu}^{(K)}(\gamma) D_{\mu\lambda}^{(K)}(\gamma_j) T_{aa'}^j e^{-i\phi_j} D_{\nu'\lambda'}^{(K')*}(\gamma) + \mathrm{h.c.} \\
&= -\delta_{KK'} \sum_{j=1}^{2g} D_{\lambda'\lambda}^{(K)}(\gamma_j) \delta_{\nu\nu'} T_{aa'}^j e^{-i\phi_j} + \mathrm{h.c.} \quad (A.2)
\end{aligned}
$$

In the last step, we have used the Schur orthogonality relation

$$\frac{d_K}{|G|} \sum_{\gamma \in G} D_{\nu\mu}^{(K)}(\gamma) D_{\nu'\lambda'}^{(K')*}(\gamma) = \delta_{KK'} \delta_{\nu\nu'} \delta_{\mu\lambda'}, \quad (A.3)$$

which applies for any two irreps $K$ and $K'$. Combining Eqs. (A.1) and (A.2), we obtain Eqs. (9) and (10) in the main text.

# B  Fourier transform on hyperbolic periodic clusters

In this appendix, we elaborate on the hyperbolic Fourier transform (17):

$$f(\gamma) = \frac{1}{|G|} \sum_{K \in \mathrm{BZ}(G)} \sum_{\lambda, \nu = 1}^{d_K} d_K f_{\lambda\nu}^{(K)} D_{\nu\lambda}^{(K)*}(\gamma), \qquad f_{\lambda\nu}^{(K)} = \sum_{\gamma \in G} f(\gamma) D_{\nu\lambda}^{(K)}(\gamma). \tag{B.1}$$

In particular, we prove that the transforms are indeed inverses of each other. Then, we introduce the Fourier transform of a matrix kernel and narrow attention to the translationally invariant case, which is relevant for the conductivity tensor. Finally, we show that the generalized Fourier transform satisfies a convolution theorem.

The transforms in (B.1) are inverses of each other, which can be readily verified using the following orthogonality relations:

$$\frac{d_K}{|G|} \sum_{\gamma \in G} D_{\nu\lambda}^{(K)*}(\gamma) D_{\nu'\lambda'}^{(K')}(\gamma) = \delta_{KK'} \delta_{\nu\nu'} \delta_{\lambda\lambda'}, \qquad \frac{1}{|G|} \sum_{K \in \mathrm{BZ}(G)} \sum_{\lambda, \nu = 1}^{d_K} d_K D_{\nu\lambda}^{(K)*}(\gamma) D_{\nu\lambda}^{(K)}(\gamma') = \delta_{\gamma,\gamma'}. \tag{B.2}$$

When $G = \mathbb{Z}_N$, we recover the standard Euclidean orthogonality relations

$$\frac{1}{N} \sum_{x} e^{i(k_n - k_m)x} = \delta_{k_n, k_m}, \qquad \frac{1}{N} \sum_{k_n} e^{ik_n(x-x')} = \delta_{x,x'}. \tag{B.3}$$

The first equation in (B.2) is precisely the Schur orthogonality relation, already stated as Eq. (A.3), and the second equation follows from combining the unitarity property

$$D_{\nu\lambda}^{(K)*}(\gamma) = D_{\lambda\nu}^{(K)}(\gamma^{-1}), \tag{B.4}$$

with the second orthogonality relation for characters (columns of character table are orthogonal),

$$\frac{1}{|G|} \sum_{K \in \mathrm{BZ}(G)} \chi^{(K)*}(\gamma) \chi^{(K)}(\gamma') = \begin{cases} 1/n_c(\gamma), & \gamma \sim \gamma', \\ 0, & \text{otherwise,} \end{cases} \tag{B.5}$$

where $\chi^{(K)}(\gamma) = \mathrm{Tr}\, D^{(K)}(\gamma)$ is the character of $\gamma$ in the irrep $K$. Here $\gamma \sim \gamma'$ iff $\gamma$ and $\gamma'$ belong to the same conjugacy class, and $n_c(\gamma)$ is the cardinality of the conjugacy class to which $\gamma$ belongs. To prove the second equation in (B.2), we rewrite the left-hand side as

$$\frac{1}{|G|} \sum_{K \in \mathrm{BZ}(G)} \sum_{\lambda, \nu = 1}^{d_K} d_K D_{\nu\lambda}^{(K)*}(\gamma) D_{\nu\lambda}^{(K)}(\gamma') = \frac{1}{|G|} \sum_{K \in \mathrm{BZ}(G)} \sum_{\lambda, \nu = 1}^{d_K} d_K D_{\lambda\nu}^{(K)}(\gamma^{-1}) D_{\nu\lambda}^{(K)}(\gamma')$$

$$= \frac{1}{|G|} \sum_{K \in \mathrm{BZ}(G)} \sum_{\lambda = 1}^{d_K} d_K D_{\lambda\lambda}^{(K)}(\gamma^{-1}\gamma')$$

$$= \frac{1}{|G|} \sum_{K \in \mathrm{BZ}(G)} d_K \chi^{(K)}(\gamma^{-1}\gamma'). \tag{B.6}$$

We now apply Eq. (B.5) by noticing that $\chi^{(K)}(e) = d_K = \chi^{(K)*}(e)$, where $e$ is the identity element, and $e$ is the only element in its conjugacy class. This gives the stated result.

The Fourier transform of a matrix kernel $h(\gamma, \gamma')$ can be similarly defined:

$$h(\gamma, \gamma') = \frac{1}{|G|^2} \sum_{K, K' \in \mathrm{BZ}(G)} \sum_{\lambda, \nu = 1}^{d_K} \sum_{\lambda', \nu' = 1}^{d_{K'}} d_K d_{K'} D_{\nu\lambda}^{(K)*}(\gamma) h_{\lambda\nu,\lambda'\nu'}^{(K,K')} D_{\nu'\lambda'}^{(K')}(\gamma'), \tag{B.7}$$

with inverse

$$h_{\lambda\nu,\lambda'\nu'}^{(K,K')} = \sum_{\gamma,\gamma'\in G} D_{\nu\lambda}^{(K)}(\gamma)h(\gamma,\gamma')D_{\nu'\lambda'}^{(K')*}(\gamma').\tag{B.8}$$

In the presence of translational symmetry, the Fourier transform can be greatly simplified. By translational symmetry, we mean $h(\gamma,\gamma') = h(\tilde\gamma\gamma,\tilde\gamma\gamma')$ for all $\tilde\gamma\in G$. In this case,

$$
\begin{aligned}
h(\gamma,\gamma') &= \frac{1}{|G|}\sum_{\tilde\gamma\in G}h(\tilde\gamma\gamma,\tilde\gamma\gamma')\\
&= \frac{1}{|G|^3}\sum_{\tilde\gamma\in G}\sum_{K,K'\in\mathrm{BZ}(G)}\sum_{\lambda,\nu=1}^{d_K}\sum_{\lambda',\nu'=1}^{d_{K'}}d_K d_{K'}D_{\nu\lambda}^{(K)*}(\tilde\gamma\gamma)h_{\lambda\nu,\lambda'\nu'}^{(K,K')}D_{\nu'\lambda'}^{(K')}(\tilde\gamma\gamma')\\
&= \frac{1}{|G|^3}\sum_{\tilde\gamma\in G}\sum_{K,K'\in\mathrm{BZ}(G)}\sum_{\lambda,\nu=1}^{d_K}\sum_{\lambda',\nu'=1}^{d_{K'}}d_K d_{K'}D_{\nu\lambda}^{(K)*}(\tilde\gamma\gamma'^{-1}\gamma)h_{\lambda\nu,\lambda'\nu'}^{(K,K')}D_{\nu'\lambda'}^{(K')}(\tilde\gamma)\\
&= \frac{1}{|G|^3}\sum_{\tilde\gamma\in G}\sum_{K,K'\in\mathrm{BZ}(G)}\sum_{\lambda,\nu,\mu=1}^{d_K}\sum_{\lambda',\nu'=1}^{d_{K'}}d_K d_{K'}D_{\nu\mu}^{(K)*}(\tilde\gamma)D_{\mu\lambda}^{(K)*}(\gamma'^{-1}\gamma)h_{\lambda\nu,\lambda'\nu'}^{(K,K')}D_{\nu'\lambda'}^{(K')}(\tilde\gamma)\\
&= \frac{1}{|G|^2}\sum_{K\in\mathrm{BZ}(G)}\sum_{\lambda,\lambda',\nu=1}^{d_K}d_K h_{\lambda\nu,\lambda'\nu}^{(K,K)}D_{\lambda'\lambda}^{(K)*}(\gamma'^{-1}\gamma).
\end{aligned}\tag{B.9}
$$

In the last step we have used the Schur orthogonality theorem. If we now define

$$h_{\lambda\lambda'}^{(K)} \equiv \frac{1}{|G|}\sum_{\nu=1}^{d_K}h_{\lambda\nu,\lambda'\nu}^{(K,K)},\tag{B.10}$$

we obtain

$$h(\gamma,\gamma') = \frac{1}{|G|}\sum_{K\in\mathrm{BZ}(G)}\sum_{\lambda,\lambda'=1}^{d_K}d_K h_{\lambda\lambda'}^{(K)}D_{\lambda'\lambda}^{(K)*}(\gamma'^{-1}\gamma),\tag{B.11}$$

which is simply the Fourier transform with respect to the single "difference variable" $\gamma'^{-1}\gamma$. In particular, this tells us that $h(\gamma,\gamma') = h(\gamma'^{-1}\gamma)$ depends only on group elements through $\gamma'^{-1}\gamma$. In analogy with Eqs. (B.1), we can deduce the inverse transform to be

$$h_{\lambda\lambda'}^{(K)} = \sum_{\gamma\in G}h(\gamma)D_{\lambda'\lambda}^{(K)}(\gamma).\tag{B.12}$$

Just as for the Euclidean Fourier transform, the hyperbolic Fourier transform also obeys a convolution theorem. For a general finite non-Abelian group, the convolution of two functions $h$ and $g$ is defined as

$$f(\gamma) = \sum_{\gamma'\in G}h(\gamma'^{-1}\gamma)g(\gamma').\tag{B.13}$$

Taking the Fourier transform,

$$
\begin{aligned}
f_{\lambda\nu}^{(K)} &= \sum_{\gamma\in G}f(\gamma)D_{\nu\lambda}^{(K)}(\gamma) = \sum_{\gamma,\gamma'\in G}h(\gamma'^{-1}\gamma)g(\gamma')D_{\nu\lambda}^{(K)}(\gamma) = \sum_{\gamma,\gamma'\in G}h(\gamma'^{-1}\gamma)g(\gamma')D_{\nu\lambda}^{(K)}(\gamma'\gamma'^{-1}\gamma)\\
&= \sum_{\gamma,\gamma'\in G}\sum_{\mu=1}^{d_K}h(\gamma'^{-1}\gamma)D_{\mu\lambda}^{(K)}(\gamma'^{-1}\gamma)g(\gamma')D_{\nu\mu}^{(K)}(\gamma') = \sum_{\gamma,\gamma'\in G}\sum_{\mu=1}^{d_K}h(\gamma)D_{\mu\lambda}^{(K)}(\gamma)g(\gamma')D_{\nu\mu}^{(K)}(\gamma')\\
&= \sum_{\mu=1}^{d_K}h_{\lambda\mu}^{(K)}g_{\mu\nu}^{(K)},
\end{aligned}\tag{B.14}
$$

which is Eq. (21) in the main text.

## C Kubo formula and the Berry curvature

In this appendix, we provide a detailed derivation of the Hall conductivity in Eq. (24). We take the Fourier transform of the Kubo formula and the current operator. Combining the two, we show that the Hall conductivity in the uniform limit is determined by the Berry curvature.

First, assuming translational invariance, we take the Fourier transform of the conductivity. Our starting point is the Kubo formula in Eq. (15). Taking the Fourier transform using Eq. (B.8), we obtain

$$
\begin{aligned}
\sigma^{(Q,Q')}_{ij;\tilde{\lambda}\tilde{\nu},\tilde{\lambda}'\tilde{\nu}'} &= -i\hbar \sum_{\Omega \neq \mathrm{GS}} \sum_{\gamma,\gamma' \in G} D^{(Q)}_{\tilde{\nu}\tilde{\lambda}}(\gamma) \frac{\langle \mathrm{GS}|\hat{J}_i(\gamma)|\Omega\rangle \langle \Omega|\hat{J}_j(\gamma')|\mathrm{GS}\rangle}{(E_\Omega - E_{\mathrm{GS}})^2} D^{(Q')*}_{\tilde{\nu}'\tilde{\lambda}'}(\gamma') - (i \leftrightarrow j) \\
&= -i\hbar \sum_{\Omega \neq \mathrm{GS}} \frac{\langle \mathrm{GS}|\hat{J}^{(Q)}_{i;\tilde{\lambda}\tilde{\nu}}|\Omega\rangle \langle \Omega|\hat{J}^{(-Q')}_{j;\tilde{\lambda}'\tilde{\nu}'}|\mathrm{GS}\rangle}{(E_\Omega - E_{\mathrm{GS}})^2} - (i \leftrightarrow j),
\end{aligned}
\tag{C.1}
$$

where $\hat{J}^{(Q)}_{i;\tilde{\lambda}\tilde{\nu}}$ is the Fourier transform of the current operator and we denote by $-Q'$ the complex conjugate representation of $Q'$, i.e.,

$$
D^{(-Q)}_{\nu\lambda}(\gamma) = D^{(Q)*}_{\nu\lambda}(\gamma).
\tag{C.2}
$$

By translational symmetry, not all Fourier coefficients are independent. The independent components are the linear combinations defined in Eq. (B.10), which for the conductivity are given by

$$
\sigma^{(Q)}_{ij;\tilde{\lambda}\tilde{\lambda}'} = -i\hbar \frac{1}{|G|} \sum_{\Omega \neq \mathrm{GS}} \sum_{\tilde{\nu}=1}^{d_K} \frac{\langle \mathrm{GS}|\hat{J}^{(Q)}_{i;\tilde{\lambda}\tilde{\nu}}|\Omega\rangle \langle \Omega|\hat{J}^{(-Q)}_{j;\tilde{\lambda}'\tilde{\nu}}|\mathrm{GS}\rangle}{(E_\Omega - E_{\mathrm{GS}})^2} - (i \leftrightarrow j).
\tag{C.3}
$$

To proceed, we need the Fourier transform of the current operator. Using the definition in Eq. (12) and with the Hamiltonian in Eq. (3), we have the following explicit form for the current operator:

$$
\hat{J}_j(\gamma) = -\frac{2\pi i}{\Phi_0} \sum_{a,a'=1}^{N_s} T^j_{aa'} e^{-i\phi_j} \hat{c}^\dagger_{\gamma\gamma_j,a} \hat{c}_{\gamma,a'} + \mathrm{h.c.}
\tag{C.4}
$$

With translational symmetry, it is convenient to pass to the irrep basis. Although in the main text the irrep basis was introduced in the first-quantized setting, it can be straight-forwardly generalized to second-quantized operators. The electron creation operators in the lattice basis are defined as $\hat{c}^\dagger_{\gamma,a}|0\rangle = |\gamma,a\rangle$. They satisfy the anticommutation relation $\{\hat{c}_{\gamma,a}, \hat{c}^\dagger_{\gamma',a'}\} = \delta_{\gamma,\gamma'}\delta_{aa'}$. In the main text, we have also introduced the irrep basis $|K,\lambda,\nu;a\rangle$. We define the analogous fermion operators through $\hat{c}^{(K)\dagger}_{\lambda\nu,a}|0\rangle \equiv |K,\lambda,\nu;a\rangle$. Using Eq. (6), the two sets of operators are related by the change of basis

$$
\hat{c}^\dagger_{\gamma,a} = \sum_{K \in \mathrm{BZ}(G)} \sum_{\lambda,\nu=1}^{d_K} \hat{c}^{(K)\dagger}_{\lambda\nu,a} \sqrt{\frac{d_K}{|G|}} D^{(K)}_{\nu\lambda}(\gamma), \qquad \hat{c}^{(K)\dagger}_{\lambda\nu,a} = \sum_{\gamma \in G} \hat{c}^\dagger_{\gamma,a} \sqrt{\frac{d_K}{|G|}} D^{(K)*}_{\nu\lambda}(\gamma).
\tag{C.5}
$$

As the change of basis is unitary, the anticommutation relations are preserved. In particular, $\left\{\hat{c}^{(K)}_{\lambda\nu,a},\hat{c}^{(K')\dagger}_{\lambda'\nu',a'}\right\}=\delta_{KK'}\delta_{\lambda\lambda'}\delta_{\nu\nu'}\delta_{aa'}$. Passing to the irrep basis, the current operator becomes

$$
\begin{aligned}
\hat{J}_j(\gamma) &= -\frac{2\pi i}{\Phi_0}\sum_{a,a'=1}^{N_s}T^j_{aa'}e^{-i\phi_j}\hat{c}^\dagger_{\gamma\gamma_j,a}\hat{c}_{\gamma,a'}+\text{h.c.} \\
&= -\frac{2\pi i}{\Phi_0}\sum_{K,K'\in\text{BZ}(G)}\sum_{\lambda,\nu=1}^{d_K}\sum_{\lambda',\nu'=1}^{d_{K'}}\sum_{a,a'=1}^{N_s}\frac{\sqrt{d_K d'_K}}{|G|}T^j_{aa'}e^{-i\phi_j}D^{(K)}_{\nu\lambda}(\gamma\gamma_j)D^{(K')*}_{\nu'\lambda'}(\gamma)\hat{c}^{(K)\dagger}_{\lambda\nu,a}\hat{c}^{(K')}_{\lambda'\nu',a'}+\text{h.c.} \\
&= -\frac{2\pi i}{\Phi_0}\sum_{K,K'\in\text{BZ}(G)}\sum_{\lambda,\nu,\mu=1}^{d_K}\sum_{\lambda',\nu'=1}^{d_{K'}}\sum_{a,a'=1}^{N_s}\frac{\sqrt{d_K d'_K}}{|G|}T^j_{aa'}e^{-i\phi_j}D^{(K)}_{\nu\mu}(\gamma)D^{(K)}_{\mu\lambda}(\gamma_j)D^{(K')*}_{\nu'\lambda'}(\gamma)\hat{c}^{(K)\dagger}_{\lambda\nu,a}\hat{c}^{(K')}_{\lambda'\nu',a'}+\text{h.c.} \\
&= -\frac{2\pi}{\Phi_0}\sum_{K,K'\in\text{BZ}(G)}\sum_{\lambda,\nu,\mu=1}^{d_K}\sum_{\lambda',\nu'=1}^{d_{K'}}\sum_{a,a'=1}^{N_s}\frac{\sqrt{d_K d'_K}}{|G|}\partial_{\phi_j}H^{(K)}_{\lambda a,\mu a'}D^{(K)}_{\nu\mu}(\gamma)D^{(K')*}_{\nu'\lambda'}(\gamma)\hat{c}^{(K)\dagger}_{\lambda\nu,a}\hat{c}^{(K')}_{\lambda'\nu',a'}.
\end{aligned}
\tag{C.6}
$$

For notational convenience, we have introduced

$$
H^{(K)}_{\lambda a,\mu a'}=H^{(K)}_{\lambda\sigma a,\mu\sigma a'}=\frac{1}{d_K}\sum_{\sigma=1}^{d_K}H^{(K)}_{\lambda\sigma a,\mu\sigma a'},
\tag{C.7}
$$

where $H^{(K)}_{\lambda\sigma a,\mu\sigma a'}$ is the Bloch Hamiltonian defined in Eq. (10). The second equality follows because $H_{\lambda\sigma a,\mu\sigma a'}$ is independent of $\sigma$; therefore, tracing out the $\sigma$ index cancels with the $d_K$ in the denominator. Taking the Fourier transform, we get

$$
\hat{J}^{(Q)}_{j;\tilde{\lambda}\tilde{\nu}}=-\frac{2\pi}{\Phi_0}\sum_{K,K'\in\text{BZ}(G)}\sum_{\lambda,\nu,\mu=1}^{d_K}\sum_{\lambda',\nu'=1}^{d_{K'}}\sum_{a,a'=1}^{N_s}\sqrt{\frac{d_{K'}}{d_K}}\partial_{\phi_j}H^{(K)}_{\lambda a,\mu a'}C^{(K',-Q,K)}_{\lambda'\nu',\tilde{\lambda}\tilde{\nu},\mu\nu}\hat{c}^{(K)\dagger}_{\lambda\nu,a}\hat{c}^{(K')}_{\lambda'\nu',a'},
\tag{C.8}
$$

where

$$
C^{(K',-Q,K)}_{\lambda'\nu',\tilde{\lambda}\tilde{\nu},\mu\nu}=\frac{d_K}{|G|}\sum_{\gamma\in G}D^{(K)}_{\nu\mu}(\gamma)D^{(K')*}_{\nu'\lambda'}(\gamma)D^{(-Q)*}_{\tilde{\nu}\tilde{\lambda}}(\gamma).
\tag{C.9}
$$

The coefficient $C^{(K',-Q,K)}_{\lambda'\nu',\tilde{\lambda}\tilde{\nu},\mu\nu}$ can be expressed in terms of Clebsch-Gordan coefficients for the group $G$. Consider the tensor product space $\mathcal{H}^{(K')}_{\rho'}\otimes\mathcal{H}^{(-Q)}_{\tilde{\rho}}$, where $\mathcal{H}^{(K')}_{\rho'}=\text{Span}\{|K',\rho',\lambda'\rangle:\lambda'=1,\ldots,d_{K'}\}$ and $\mathcal{H}^{(-Q)}_{\tilde{\rho}}=\text{Span}\{|-Q,\tilde{\rho},\tilde{\lambda}\rangle:\tilde{\lambda}=1,\ldots,d_{-Q}\}$ for some choice of $\rho'=1,\ldots,d_{K'}$ and $\tilde{\rho}=1,\ldots,d_{-Q}$. It has, as a choice, the basis set

$$
\{|K',\rho',\lambda';-Q,\tilde{\rho},\tilde{\lambda}\rangle\equiv|K',\rho',\lambda'\rangle\otimes|-Q,\tilde{\rho},\tilde{\lambda}\rangle:\lambda'=1,\ldots,d_{K'};\tilde{\lambda}=1,\ldots,d_{-Q}\}.
\tag{C.10}
$$

This space transforms in the $D^{(K')}\otimes D^{(-Q)}$ representation, which is in general reducible. It would be more convenient to switch to the basis, which we denote $|K,\rho,\mu\rangle$, that brings the representation matrices to block diagonal form. Here $K\in K'\otimes(-Q)$ is an irrep that appears in the direct-sum decomposition of the tensor product, $\mu=1,\ldots,d_K$ labels the vector within the irrep, and $\rho=1,\ldots,m_K$ is an index that accounts for the multiplicity. Note that, unlike before, the multiplicity index does not in general run from 1 to $d_K$ because the tensor product space does not transform in the regular representation. The two basis sets are related by

$$
|K',\rho',\lambda';-Q,\tilde{\rho},\tilde{\lambda}\rangle=\sum_{K\in K'\otimes(-Q)}\sum_{\rho=1}^{m_K}\sum_{\mu=1}^{d_K}|K,\rho,\mu\rangle\langle K,\rho,\mu|K',\rho',\lambda';-Q,\tilde{\rho},\tilde{\lambda}\rangle,
\tag{C.11}
$$

where $\langle K, \rho, \lambda | K', \rho', \lambda'; -Q, \tilde{\rho}, \tilde{\lambda} \rangle$ is a Clebsch-Gordan coefficient. Applying the group transformation to Eq. (C.11), we obtain

$$
\sum_{\nu'=1}^{d_{K'}} \sum_{\tilde{\nu}=1}^{d_{-Q}} |K', \rho', \nu'; -Q, \tilde{\rho}, \tilde{\nu}\rangle D_{\nu'\lambda'}^{(K')}(\gamma) D_{\tilde{\nu}\tilde{\lambda}}^{(-Q)}(\gamma)
$$
$$
= \sum_{K \in K' \otimes (-Q)} \sum_{\rho=1}^{m_K} \sum_{\mu,\nu=1}^{d_K} |K, \rho, \nu\rangle D_{\nu\mu}^{(K)}(\gamma) \langle K, \rho, \mu | K', \rho', \lambda'; -Q, \tilde{\rho}, \tilde{\lambda}\rangle . \tag{C.12}
$$

Applying the orthogonality theorem and using that $|K', \rho', \nu'; -Q, \tilde{\lambda}, \tilde{\nu}\rangle$ forms a basis, we obtain an expression for the coefficient $C_{\lambda'\nu',\tilde{\lambda}\tilde{\nu},\mu\nu}^{(K',-Q,K)}$ in terms of Clebsch-Gordan coefficients:

$$
C_{\lambda'\nu',\tilde{\lambda}\tilde{\nu},\mu\nu}^{(K',-Q,K)} = \sum_{\rho=1}^{m_K} \langle K', \rho', \lambda'; -Q, \tilde{\rho}, \tilde{\lambda} | K, \rho, \mu \rangle \langle K, \rho, \nu | K', \rho', \nu'; -Q, \tilde{\rho}, \tilde{\nu} \rangle . \tag{C.13}
$$

Note that the coefficient is independent of our choice of $\rho'$ and $\tilde{\rho}$.

The expression in Eq. (C.13) gives a simple interpretation for the hyperbolic current operator in Eq. (C.8). For illustrative purposes, consider again the case of a one-dimensional chain, in which $G = \mathbb{Z}_N$. The coefficient in Eq. (C.9) becomes

$$
C^{(k',-q,k)} = \frac{1}{N} \sum_x e^{-i(k-k'+q)x} = \delta_{k,k'-q} . \tag{C.14}
$$

We have suppressed the $\nu$ and $\lambda$ subscripts because they all take only one value. The coefficient $C^{(k',-q,k)}$ imposes the selection rule $k = k'-q$, which is the conservation of crystal momentum [see Fig. 6(a)]. In this case, we recover the usual current operator

$$
\hat{j}_j^{(q)} = -\frac{2\pi}{\Phi_0} \sum_k \partial_{k_j} H_{aa'}^{(k-q)} \hat{c}_a^{(k-q)\dagger} \hat{c}_{a'}^{(k)} . \tag{C.15}
$$

Here we have used that threading the flux $\phi_j$ in Euclidean space changes the wavevector as $k_j \mapsto k_j + \phi_j$ to change the derivative $\partial_{\phi_j}$ to $\partial_{k_j}$. The case of a non-Abelian group $G$ is analogous. The process of addition of crystal momentum is replaced with the tensor product of two irreps. The selection rule which imposes conservation of momentum is replaced with Clebsch-Gordan coefficients in Eq. (C.13) [see Fig. 6(b)].

For our purposes, we are interested in the uniform limit, i.e., $Q$ is the trivial representation (denoted "0"), in which case we can straightforwardly apply the orthogonality theorem to Eq. (C.9) to obtain the simple expression

$$
C_{\lambda'\nu';00;\mu\nu}^{(K',0,K)} = \delta_{KK'} \delta_{\lambda'\mu} \delta_{\nu'\nu} . \tag{C.16}
$$

This gives the current operator in the uniform limit

$$
\hat{j}_j^{(0)} = -\frac{2\pi}{\Phi_0} \sum_{K \in \text{BZ}(G)} \sum_{\lambda,\lambda',\nu=1}^{d_K} \sum_{a,a'=1}^{N_s} \partial_{\phi_j} H_{\lambda a,\lambda' a'}^{(K)} \hat{c}_{\lambda\nu,a}^{(K)\dagger} \hat{c}_{\lambda'\nu,a'}^{(K)} . \tag{C.17}
$$

To evaluate matrix elements, it is convenient to switch to the band basis, a basis that diagonalizes the Hamiltonian. We define the fermion operators $\hat{c}_{n\nu}^{(K)\dagger} |0\rangle = |\psi_{n\nu}^{(K)}\rangle$. They are related to the operators $\hat{c}_{\lambda,\nu;a}^{(K)\dagger}$ through the change of basis

$$
\hat{c}_{n\nu}^{(K)\dagger} = \sum_{K \in \text{BZ}(G)} \sum_{\lambda,\nu=1}^{d_K} \sum_{a=1}^{N_s} \hat{c}_{\lambda\nu,a}^{(K)\dagger} \langle K, \lambda, \nu; a | \psi_{n\nu}^{(K)} \rangle , \qquad \hat{c}_{\lambda\nu,a}^{(K)\dagger} = \sum_{n=1}^{d_K N_s} \hat{c}_{n\nu}^{(K)\dagger} \langle \psi_{n\nu}^{(K)} | K, \lambda, \nu; a \rangle . \tag{C.18}
$$

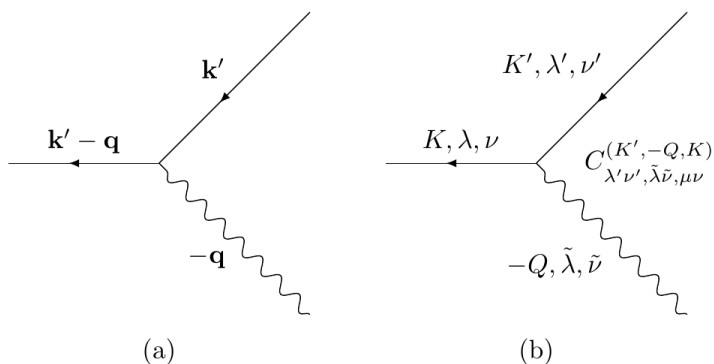

Figure 6: (a) Depiction of the conservation of crystal wavevector in Euclidean lattices. An electron with wavevector $\mathbf{k}'$ is annihilated and one with $\mathbf{k} = \mathbf{k}' - \mathbf{q}$ is created. (b) On a hyperbolic lattice, the wavevector is replaced by the tuple $K', \lambda', \nu'$ and the addition of wavevector translates to tensor product of irreps. The selection rules are dictated by the coefficient $C^{(K', -Q, K)}_{\lambda' \nu', \tilde{\lambda} \tilde{\nu}, \mu \nu}$.

As the change of basis is unitary, the new operators preserve the canonical anticommutation relations. In particular, the non-trivial relation is $\left\{ \hat{c}^{(K)}_{n\nu}, \hat{c}^{(K')\dagger}_{m\mu} \right\} = \delta_{KK'} \delta_{nm} \delta_{\nu\mu}$. In the band basis, the current operator becomes

$$\hat{J}^{(0)}_j = -\frac{2\pi}{\Phi_0} \sum_{K \in \mathrm{BZ}(G)} \sum_{n,m=1}^{N_s} \sum_{\nu=1}^{d_K} \langle \psi^{(K)}_{n\nu} | \partial_{\phi_j} \hat{H}_1 | \psi^{(K)}_{m\nu} \rangle \hat{c}^{(K)\dagger}_{n\nu} \hat{c}^{(K)}_{m\nu}. \tag{C.19}$$

Using this expression, we can evaluate the matrix elements in Eq. (C.3). For non-interacting electrons,

$$\langle \mathrm{GS} | \hat{c}^{(K)\dagger}_{n\nu} \hat{c}^{(K)}_{m\nu} | \Omega \rangle \langle \Omega | \hat{c}^{(K')\dagger}_{n'\mu} \hat{c}^{(K')}_{m'\mu} | \mathrm{GS} \rangle = \delta_{KK'} \delta_{nm'} \delta_{mn'} \delta_{\nu\mu} n_\mathrm{F}(\xi^{(K)}_n)[1 - n_\mathrm{F}(\xi^{(K)}_m)], \tag{C.20}$$

where $\xi^{(K)}_n = E^{(K)}_n - E_F$ with $E^{(K)}_n$ the Bloch energies satisfying $\hat{H}_1 | \psi^{(K)}_{n\nu} \rangle = E^{(K)}_n | \psi^{(K)}_{n\nu} \rangle$, and $n_\mathrm{F}(\xi) = \theta(-\xi)$ is a ground-state occupation factor with $\theta(x)$ the Heaviside step function. Inserting this relation into Eq. (C.3), we obtain

$$\sigma^{(0)}_{ij} = -\frac{e^2}{h} \frac{2\pi i}{|G|} \sum_{K \in \mathrm{BZ}(G)} \sum_{n<0, m>0} \sum_{\nu=1}^{d_K} \frac{\langle \psi^{(K)}_{n\nu} | \partial_{\phi_i} \hat{H}_1 | \psi^{(K)}_{m\nu} \rangle \langle \psi^{(K)}_{m\nu} | \partial_{\phi_j} \hat{H}_1 | \psi^{(K)}_{n\nu} \rangle}{(\xi^{(K)}_n - \xi^{(K)}_m)^2} - (i \leftrightarrow j). \tag{C.21}$$

Here $n < 0$ denotes the filled states and $m > 0$ the empty ones.

We now make the connection between the Hall conductivity and Berry curvature. The Berry curvature of all the filled states is defined as

$$F_{ij} = i \sum_{K \in \mathrm{BZ}(G)} \sum_{n<0} \sum_{\nu=1}^{d_K} \langle \partial_{\phi_i} u^{(K)}_{n\nu} | \partial_{\phi_j} u^{(K)}_{n\nu} \rangle - (i \leftrightarrow j)$$

$$= i \sum_{K \in \mathrm{BZ}(G)} \sum_{n<0} \sum_{\nu=1}^{d_K} \langle \partial_{\phi_i} \psi^{(K)}_{n\nu} | \partial_{\phi_j} \psi^{(K)}_{n\nu} \rangle - (i \leftrightarrow j). \tag{C.22}$$

We can recast it into a form that resembles Eq. (C.21). Inserting the resolution of identity $\mathbb{1}_{\mathcal{H}} = \sum_{K' \in \mathrm{BZ}(K)} \sum_{m=1}^{d_{K'} N_s} \sum_{\mu=1}^{d_{K'}} | \psi^{(K')}_{m\mu} \rangle \langle \psi^{(K')}_{m\mu} |$, we have

$$F_{ij} = i \sum_{K, K' \in \mathrm{BZ}(G)} \sum_{n<0, m>0} \sum_{\nu=1}^{d_K} \sum_{\mu=1}^{d_{K'}} \langle \partial_{\phi_i} \psi^{(K)}_{n\nu} | \psi^{(K')}_{m\mu} \rangle \langle \psi^{(K')}_{m\mu} | \partial_{\phi_j} \psi^{(K)}_{n\nu} \rangle - (i \leftrightarrow j). \tag{C.23}$$

We have only included the $m > 0$ terms because the $m < 0$ terms cancel when antisymmetrized. As $|\psi_{m\nu}^{(K')}\rangle$ and $|\psi_{n\nu}^{(K)}\rangle$ are orthonormal, for $m \neq n$,

$$
\begin{aligned}
0 &= \partial_{\phi_j} \left( \xi_n^{(K)} \langle \psi_{m\mu}^{(K')} | \psi_{n\nu}^{(K)} \rangle \right) \\
&= \partial_{\phi_j} \left( \langle \psi_{m\mu}^{(K')} | \hat{H}_1 | \psi_{n\nu}^{(K)} \rangle \right) \\
&= \xi_n^{(K)} \langle \partial_{\phi_j} \psi_{m\mu}^{(K')} | \psi_{n\nu}^{(K)} \rangle + \xi_m^{(K')} \langle \psi_{m\mu}^{(K')} | \partial_{\phi_j} \psi_{n\nu}^{(K)} \rangle + \langle \psi_{m\mu}^{(K')} | \partial_{\phi_j} \hat{H}_1 | \psi_{n\nu}^{(K)} \rangle .
\end{aligned}
\tag{C.24}
$$

From Eq. (9), we observe that $\partial_{\phi_i} \hat{H}_1$ cannot change the quantum numbers $K$ and $\mu$, which leads to the identity

$$
\langle \psi_{m\mu}^{(K')} | \partial_{\phi_j} \psi_{n\nu}^{(K)} \rangle = \frac{\langle \psi_{m\nu}^{(K')} | \partial_{\phi_j} \hat{H}_1 | \psi_{n\nu}^{(K)} \rangle}{\xi_n^{(K)} - \xi_m^{(K)}} \delta_{KK'} \delta_{\nu\mu} ,
\tag{C.25}
$$

for $n \neq m$. Consequently, we obtain

$$
F_{ij} = i \sum_{K \in \mathrm{BZ}(G)} \sum_{n<0, m>0} \sum_{\nu=1}^{d_K} \frac{\langle \psi_{n\nu}^{(K)} | \partial_{\phi_i} \hat{H}_1 | \psi_{m\nu}^{(K)} \rangle \langle \psi_{m\nu}^{(K)} | \partial_{\phi_j} \hat{H}_1 | \psi_{n\mu}^{(K)} \rangle}{(\xi_n^{(K)} - \xi_m^{(K)})^2} - (i \leftrightarrow j) .
\tag{C.26}
$$

Combining with the expression for the Hall conductivity Eq. (C.21), we obtain Eq. (24) of the main text.

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
