# Peer review of "Topological linear response of hyperbolic Chern insulators"

_SciPost Physics, doi:SciPost Phys. 17, 124 (2024)_

## Round 1 · Referee Report · Anonymous (Referee 1) · 2024-8-25

Report

The authors of the present work consider the problem of calculating the characterization of the bands in a hyperbolic lattice in terms of Chern numbers. This is a rather important task given the recent developments regarding hyperbolic crystallography. By applying the theory of representations of finite groups to the translational group on the hyperbolic lattice the authors show that the Chern number characterizing a set of bands separated by a gap is given by a band transforming under a trivial representation. In other words, the Chern number of a set of bands is completely determined by this topological invariant for the band in the trivial representation. This is the central result of the paper, which is presented in eq. 34. All other bands, even though they contribute to this topological invariant, have a contribution that is proportional to the corresponding trivial representation. The correspondence can be established, as the authors show, by a smooth mapping in the space of the irreducible representations of the hyperbolic translational group, which is a crucial step in the proof. Besides, the authors present a rather detailed account of the generalization of the usual Fourier transform in terms of the irreps of the translational group on a hyperbolic space and the generalization of the usual Kubo formula to the case of the hyperbolic lattices, where in general translations do not commute. Finally, the obtained form of the Chern number is applied to calculate the Hall conductivity in Haldane model on the hyperbolic {8,3} lattice, which is already discussed in Ref. 13 (Urwyler et al., Phys. Rev. Lett. 129, 246402 (2022)).

In my opinion, the paper presents a very important result that will spur further research on topological band theory in hyperbolic lattices. In particular, it will motivate further generalizations of the topological invariants in the tenfold periodic table to the hyperbolic matter and motivate experimental efforts to verify these theoretical predictions.

In light of this, I recommend the paper for publication in SciPost in its current form.

Recommendation

Publish (easily meets expectations and criteria for this Journal; among top 50%)

---

## Round 1 · Referee Report · Anonymous (Referee 2) · 2024-8-27

Strengths

  • Novelty
  • Thoroughness
  • presentation

Weaknesses

  • applicability to physical systems beyond synthetic ones

Report

In this work the authors generalize the Hall response to the hyperbolic lattice case. The paper reads well and deserves publication after appended comments/queries are addressed.

1.I am a bit confused about how the authors go from eq 1 to eq 3. I thought translations are generally defined as the torsion free subgroup. This has Abelian and non-Abelian sectors. Here only the Abelian case is important. I think this can be made clearer from Eq. 1 to Eq.4. I.e making this more self contained before coming to the specific Hamiltonian.

  1. Given Fig. 2, is there a connection to be made with homogies as flux threading?

  2. The authors say “where $C(K)_{ij}$ is the first Chern number associated with the irrep K and is quantized to integer values. I understand this from a general -Euclidean- point of view. But maybe the authors can elaborate. In the end we take the Abelian sector and are integrating over a closed manifold the eq 35 using a standard procure shows it a character but are all sectors [Abelian and non-Abelian] always orthogonal?

smaller comments; - What do the authors mean with “Hyperbolic matter is a novel form of synthetic matter” as this is clearly a lattice not a form of matter directly?

-I would strongly recommend the Abelian part in an updated title as this is important compared to the Euclidean cases

-With respect to Ref 59-64 it would be fair to acknowledge that all quantum Hall [Class A] crystalline responses as mapped to K-theory were pointed out in Physical Review X 7 (4), 041069 (2017), see also relation to irreps.

Requested changes

see report

Recommendation

Publish (easily meets expectations and criteria for this Journal; among top 50%)

  • validity: high
  • significance: high
  • originality: high
  • clarity: high
  • formatting: perfect
  • grammar: perfect

Author:  Canon Sun  on 2024-09-09  [id 4748]

(in reply to Report 2 on 2024-08-27)
Category:
answer to question

We thank the referee for their thorough review and insightful comments. Below we address each point raised by the referee.

  1. Clarification from Eq. (1) to Eq. (4):

We appreciate the referee's suggestion for clarity. To clarify, there is a distinction between the periodic cluster $G$, defined as a torsion-free normal subgroup, and the U(1) Peierls substitution, which is due to an external electric field. To go from Eq. (1) to (3), we first define a tight-binding model with nearest-neighbor hopping on lattice sites $G$. The Abelian U(1) Peierls substitution is then applied to this tight-binding Hamiltonian.

It is important to note that after the Peierls substitution, the translation group remains non-Abelian. We did not select only the Abelian part of the translation group. In the revised manuscript, we have added a sentence after Eq. (5) to emphasize this fact.

  1. Connection Between Homology and Flux Threading:

There is indeed a connection between homology and flux threading. Homology characterizes the number of "holes" in the surface on which the periodic cluster resides. Flux threading through these holes can be understood by introducing a flat gauge field $A$. Although the gauge field is flat (i.e., its exterior derivative vanishes), it belongs to a non-trivial cohomological class (i.e., it is not the exterior derivative of something). We have added a brief discussion at the end of the second paragraph on page 5 to elaborate on the connection.

More formally, it is indeed homology rather than homotopy which is relevant for flux threading, since the flux $\varphi_\alpha$ through the noncontractible cycle $g_\alpha \in \Gamma_{\text{PBC}} \cong\pi_1(\Sigma_h)$ (here, $\Sigma_h$ denotes the genus-$h$ surface in Fig. 2 and $\pi_1$ its fundamental group) only depends on the abelianization $\Gamma_\text{PBC}/[\Gamma_\text{PBC},\Gamma_\text{PBC}]$, which is isomorphic to the first homology group $H_1(\Sigma_h,\mathbb{Z})$. Indeed, $\Lambda_j(g_\alpha)$ remains unchanged if $g_\alpha$ is multiplied by an element of the commutator subgroup $[\Gamma_\text{PBC},\Gamma_\text{PBC}]$. The flux is given by integrating the gauge field one-form $A$ (which belongs to the cohomology group $H^1(\Sigma_h,\mathbb{Z})$) over a closed homology cycle in $H_1(\Sigma_h,\mathbb{Z})$.

  1. Elaboration on the Quantization of Chern Numbers:

The integer nature of $C^{(K)}_{ij}$ arises because it is the integral of the Berry curvature over a closed manifold. This is true also for non-Abelian states. To clarify, we did not selectively take the Abelian sector; rather, all states, including those in both Abelian and non-Abelian sectors, are included. The simplification Eq. (35) only occurs due to a relation between the Chern numbers. The orthogonality of the Abelian and non-Abelian sectors follows from the fact that states transforming in different irreps are necessarily orthogonal.

Smaller Comments:

In describing "hyperbolic matter as a novel form of synthetic matter," our intention is to refer to matter that exists on a hyperbolic lattice. For example, in this work, we are specifically discussing a Chern insulator living on a hyperbolic lattice.

Title Suggestion: We would like to respectfully point out that the Hall conductivity receives contributions from both Abelian and non-Abelian states. The significant simplification in Eq. (35) arises because those contributions are related to each other, not because we only take account of Abelian states. The existence of a relation between Abelian and non-Abelian Chern numbers is nontrivial and indeed one of the main results of our work. Therefore, we prefer not to specify "Abelian" in the title, as such an inclusion would result in a misleading representation of our main result.

Citation: We thank the referee for bringing our attention to Physical Review X 7 (4), 041069 (2017). We have added the suggested citation to the paper.

---

## Round 1 · Referee Report · Anonymous (Referee 3) · 2024-9-23

Strengths

The paper is well-written, the topic is an active direction of research in both theoretical and mathematical physics, and the result is relevant and important.

Weaknesses

N/A

Report

I would like the authors to address my two quick questions, otherwise I recommend for publication.

Attachment

Recommendation

Publish (surpasses expectations and criteria for this Journal; among top 10%)

  • validity: top
  • significance: top
  • originality: high
  • clarity: high
  • formatting: excellent
  • grammar: perfect

Author:  Canon Sun  on 2024-11-30  [id 5011]

(in reply to Report 3 on 2024-09-23)

We thank the referee for the intriguing questions. In Ref. [1], the author considered electrons on a lattice under a uniform magnetic field. The translation operators have to be gauged because even though the magnetic field is translationally invariant, the vector potential is not. Indeed, in the presence of a nonzero flux per unit cell, it is impossible to choose a gauge that preserves the original translation symmetry.

On the other hand, in the case we studied, there is zero net flux per unit cell and thus we $are$ able to choose a translationally invariant gauge. As a result, the translation operators remain the same as those in the absence of gauge field. If we were to apply a uniform magnetic field, however, the translation operators would indeed have to be gauged, in a similar way to Ref. [1].

In fact, hyperbolic band theory (HBT) in the presence of a magnetic field has been studied in J.~Phys.: Condens. Matter 33, 485602 (2021). It would be interesting to study the Hall response of a hyperbolic lattice under uniform magnetic field [see also Phys. Rev. Lett. 128, 166402 (2022)] by combining HBT in a magnetic field and the methods developed here. However, this goes beyond the scope of the present paper where we focus on Chern insulators (i.e., models "without Landau levels" like Haldane's 1988 honeycomb lattice model).

---

## Round 2 · Referee Report · Anonymous (Referee 1) · 2024-9-13

Report

I have no further comments on the manuscript. I have read the revised version, and I can recommend it for publication.

Recommendation

Publish (easily meets expectations and criteria for this Journal; among top 50%)

---

## Round 2 · Referee Report · Anonymous (Referee 2) · 2024-10-4

Report

I thank the authors for their revisions. I can endorse publication.

Recommendation

Publish (easily meets expectations and criteria for this Journal; among top 50%)

---

## Round 2 · Author Response

We thank the referees for their thorough review and insightful comments. Below we address each point raised by Referee 2.

  1. Clarification from Eq. (1) to Eq. (4):

We appreciate the referee's suggestion for clarity. To clarify, there is a distinction between the periodic cluster $G$, defined as a torsion-free normal subgroup, and the U(1) Peierls substitution, which is due to an external electric field. To go from Eq. (1) to (3), we first define a tight-binding model with nearest-neighbor hopping on lattice sites $G$. The Abelian U(1) Peierls substitution is then applied to this tight-binding Hamiltonian.

It is important to note that after the Peierls substitution, the translation group remains non-Abelian. We did not select only the Abelian part of the translation group. In the revised manuscript, we have added a sentence after Eq. (5) to emphasize this fact.

  1. Connection Between Homology and Flux Threading:

There is indeed a connection between homology and flux threading. Homology characterizes the number of "holes" in the surface on which the periodic cluster resides. Flux threading through these holes can be understood by introducing a flat gauge field $A$. Although the gauge field is flat (i.e., its exterior derivative vanishes), it belongs to a non-trivial cohomological class (i.e., it is not the exterior derivative of something). We have added a brief discussion at the end of the second paragraph on page 5 to elaborate on the connection.

More formally, it is indeed homology rather than homotopy which is relevant for flux threading, since the flux $\varphi_\alpha$ through the noncontractible cycle $\mathfrak{g}\alpha\in\Gamma\text{PBC}\cong\pi_1(\Sigma_h)$ (here, $\Sigma_h$ denotes the genus-$h$ surface in Fig.~2 and $\pi_1$ its fundamental group) only depends on the abelianization $\Gamma_\text{PBC}/[\Gamma_\text{PBC},\Gamma_\text{PBC}]$, which is isomorphic to the first homology group $H_1(\Sigma_h,\mathbb{Z})$. Indeed, $\Lambda_j(\mathfrak{g}_\alpha)$ remains unchanged if $\mathfrak{g}_\alpha$ is multiplied by an element of the commutator subgroup $[\Gamma_\text{PBC},\Gamma_\text{PBC}]$. The flux is given by integrating the gauge field one-form $A$ (which belongs to the cohomology group $H^1(\Sigma_h,\mathbb{Z})$) over a closed homology cycle in $H_1(\Sigma_h,\mathbb{Z})$.

  1. Elaboration on the Quantization of Chern Numbers:

The integer nature of $C^{(K)}_{ij}$ arises because it is the integral of the Berry curvature over a closed manifold. This is true also for non-Abelian states. To clarify, we did not selectively take the Abelian sector; rather, all states, including those in both Abelian and non-Abelian sectors, are included. The simplification Eq. (35) only occurs due to a relation between the Chern numbers. The orthogonality of the Abelian and non-Abelian sectors follows from the fact that states transforming in different irreps are necessarily orthogonal.

Smaller Comments:

In describing "hyperbolic matter as a novel form of synthetic matter," our intention is to refer to matter that exists on a hyperbolic lattice. For example, in this work, we are specifically discussing a Chern insulator living on a hyperbolic lattice.

Title Suggestion: We would like to respectfully point out that the Hall conductivity receives contributions from both Abelian and non-Abelian states. The significant simplification in Eq. (35) arises because those contributions are related to each other, not because we only take account of Abelian states. The existence of a relation between Abelian and non-Abelian Chern numbers is nontrivial and indeed one of the main results of our work. Therefore, we prefer not to specify "Abelian" in the title, as such an inclusion would result in a misleading representation of our main result.

Citation: We thank the referee for bringing our attention to Physical Review X 7 (4), 041069 (2017). We have added the suggested citation to the paper.

---

## Round 2 · List of Changes

• Page 5, at the second of the second paragraph. Added "Mathematically, the periodic cluster resides on a surface with non-trivial homology, which quantifies the number of "holes" the surface possesses. Flat connections corresponding to fluxes threaded through these holes belong to non-trivial cohomological classes."

  • Page 5, immediately below Eq. (5). Added "We note that even though the U(1) Peierls phase $e^{-i\phi_j}$ itself is Abelian, the translation symmetry of the Hamiltonian (5) is still noncommutative, described by the non-Abelian group $G$. To make those..."

  • Added code repository as Ref. [82]

  • Added Ref. [65], J. Kruthoff, J. de Boer, J. van Wezel, C. L. Kane and R.-J. Slager, Topological classification of crystalline insulators through band structure combinatorics, Phys. Rev. X 7, 041069 (2017)

---

## Editorial Decision

published